# Best Operating Conditions for Biogas Production in Some Simple Anaerobic Digestion Models

Tewfik Sari 

ITAP, INRAE, Institut Agro, University of Montpellier, 34196 Montpellier, France; tewfik.sari@inrae.fr

**Abstract:** We consider one-step and two-step simple models of anaerobic digestion that are able to adequately capture the main dynamical behaviour of the full anaerobic digestion model ADM1. We do not consider specific growth functions. We only require them to satisfy certain qualitative assumptions. These assumptions are satisfied for concave growth functions, but they are also satisfied for a large class of growth functions found in many applications. We consider the maximisation of the biogas production with respect to the operating parameters of the model, which are the dilution rate and the substrate input concentration. We give the best operating conditions and we describe them as a subset of the set of operating parameters. Our models incorporate biomass decay terms, corresponding to maintenance. Numerical plots with specified growth functions and biological parameters illustrate the obtained results.

**Keywords:** anaerobic digestion; biogas; chemostat; maintenance; operating diagram; optimization; productivity; stability

**MSC:** 34D20; 34H20; 65K10; 92C75

## 1. Introduction

Anaerobic digestion (AD) is a well known and established technology for treating waste in the methanisation of sewage sludge from wastewater treatment plants. AD enables the water industry to treat waste water as a resource for generating energy and recovering valuable by-products. In the context of renewable energy, it has now become an attractive alternative to fossil carbon [1]. AD is a complex biological process in which organic material is converted into biogas (methane) in an environment without oxygen [2–6]. One of its main disadvantages is its sensitivity to disturbances, which can lead to instability problems, in addition to a decrease in the biogas production rate [7]. Indeed, the conditions and technological parameters characterising the methane fermentation process include many parameters: hydraulic retention time, organic loading rate, anaerobic sludge concentration in the bioreactor, substrate dewatering, organic matter content, substrate dosage, mixing method and frequency, temperature, and many others.

When the experimenter does not have a mechanistic mathematical model of the process being studied, one method for selecting the best conditions for biogas production is to carry out multi-variant tests and select the most efficient variants and then optimise them and develop a mathematical model. This first approach is presented in [8–10]. On the other hand, when the experimenter has a model of the process being studied and knows or has identified its biological parameters, a good way to optimise biogas production is to look for the optimal flow rate of the bioreactor that produces the most biogas. This approach is presented in [11–21]. Therefore, mechanistic mathematical models are a good basis for monitoring and developing control strategies to optimise the operation of such processes. The present paper is a contribution to this second approach to the problem: we assume that we know a mechanistic mathematical model of the process and that we have already identified its biological parameters, and then we look to the best operating conditions for biogas production.

However, having a model of AD is not that easy. Indeed, the complexity of AD has motivated the development of mechanistic mathematical models, such as the widely used Anaerobic Digestion Model Model No. 1 (ADM1) [2]. This model has a large number of state variables and parameters. It is impossible to obtain an analytical characterization of the steady states and to describe the operating diagram (OD), that is to say, to identify the asymptotic behaviour of existing steady states as a function of the operating parameters (substrate inflow concentrations and dilution rate). To the author's knowledge, only numerical investigations are available [3]. Therefore, although ADM1 is a complex model that is widely accepted as a common platform for AD process modelling and simulation, it has a large number of parameters and states that hinder its analytic study. Due to the analytic intractability of the full ADM1, progress has been made towards the construction of simpler models that preserve biological meaning. The simplest model of the chemostat with only one biological reaction, where one substrate is consumed by one microorganism, is well understood [22–24]. However a one-step model is too simple to encapsulate the essence of AD.

More realistic models of AD are two-step models. An important contribution to the modelling of AD as a two-step is the model presented in [25], hereafter denoted as the AM2 model and studied in [15,26]. It has been shown that under some circumstances, this very simple two-step model is able to adequately capture the main dynamical behaviour of the full ADM1 [27,28]. AM2 is a four-dimensional system of ordinary differential equations and takes acidogenesis and methanogenesis into consideration. In the first step, the organic substrate is consumed by the acidogenic bacteria and produces a substrate, the Volatile Fatty Acids (VFA), while in the second step, the methanogenic population consumes VFA and produces biogas.

Another interesting simple AD model, with eight state variables, was considered in [21,29,30]. This model takes into consideration acidogenesis, acetogenesis, and methanogenesis. We also mention the mathematical model considered in [31], which also added the hydrolysis step in the model. It is also worth mentioning the models of AD that include the evolution of biogas and hydrogen [32–34].

The problem of optimising biogas production for one-step AD models is studied in [13,14] and for the AM2 model in [11,15–18]. This problem is also analysed in [21,31,35], where models with more steps for AD are considered.

The OD of a model has operating parameters as its coordinates, and the various regions defined within it correspond to qualitatively different asymptotic behaviours. The operating parameters are the input concentrations of substrates and the flow rate. We call them operating parameters, although they are not always under the control of the experimenter. Indeed, in most practical cases, one can at best store material upstream and control the flow rate. The concentration of the input substrate is rarely a control parameter. However, this parameter is known to the experimenter and is not of the same nature as the biological parameters, on which the experimenter can only act with great difficulty. In most of the results, we will assume that the input concentration of substrate is fixed, and we want to determine the corresponding optimal flow rate. Apart from the operating parameters, which can vary, all other parameters have biological meaning and are fitted using experimental data from ecological and/or biological observations of organisms and substrates. When the biological parameters are determined, it is then easy to plot the operating diagram and thus have a prediction of the behaviour of the system as a function of the operating parameters.

The OD is then the bifurcation diagram that shows how the system behaves when we vary the operating (control) parameters. This diagram shows how extensive the parameter region is, where some asymptotic behaviours occur. This bifurcation diagram is very useful to understand the model from both the mathematical and biological points of view. Its importance for bioreactors was emphasized in [36]. This diagram is often constructed both in the biological literature [15,29,35–38] and the mathematical literature [3,21,30,39–44].

In the present work, we consider the one-step model and the AM2 model, and we give the best operating conditions for biogas production, that is to say, we give the subset of the OD corresponding to the maximal flow rate of the biogas. This set of the best operating conditions in the OD indicates to the experimenter how to choose the operating condition such that the system produces the maximum of biogas. The surprising result for AM2 is that the optimal steady state can involve the extinction of the acidogenic bacteria [11]. This property was also observed for more complex models [21,31]. We address this problem and fully describe the operating conditions under which this situation is encountered. Another very important phenomenon, which was observed in [35], is that the best biogas produced is sometimes obtained for operating parameters for which the system has bistability. This issue is also addressed, and the set of operating parameters for which the system may be in such a situation is fully described.

The paper is organized as follows. In Section 2, we describe the one-step and two-step models of AD that are studied in this paper. We give the steady states of the models and their biogas flow rate or productivity. We state the problems of optimisation that will be considered later. The results for one-step models are given in Section 3.1. The particular case when the biomass mortality is neglected is considered in Section 3.1.8, and applications to various growth functions that were considered in the literature are given in Appendix A.5. The results for two-step models are listed in Section 3.2, and the applications of our theory to the classical AM2 model are emphasized in Section 3.2.4. We discuss and compare our results with the results of the existing literature in Section 3.3. Finally, Sections 4 and 5 draw some discussions, conclusions, and perspectives. The proofs and supplementary information are given in Appendixes A and B.

## 2. Materials and Methods

We consider a continuous stirred-tank reactor (CSTR), also called a bioreactor or a chemostat, where a single population of micro-organisms is growing on a single limiting substrate. We also consider the more complex situation where this population produces a substrate which is itself consumed by a second population. The limiting substrate is fed into the culture vessel with a constant concentration at flow rate $Q$. The culture medium is withdrawn at the same flow rate $Q$ so that the culture volume $V$ in the vessel is kept constant.

The dilution rate $D$ is defined as $D = Q/V$ and is the inverse of the residence time. We will take into account that the residence time of the liquid (culture medium) in the bioreactor may be shorter than that of the solids (micro-organisms), which is common in bioreactors.

We also take maintenance into account. Consumption of energy for all processes other than growth is called maintenance. In situations where microbial cells are located in a favourable environment, maintenance can often be neglected. In other situations, however, a significant portion of the energy-yielding substrate that could be used for growth is consumed for maintenance [45]. In the ADM1 model and also in some simple models of AD, maintenance is taken into account as decay [2,37,38,43,44].

It is assumed that the other required substrates are provided in excess, that the culture medium is perfectly mixed and that the environmental conditions (temperature and pH) are regulated at appropriate constant values.

### 2.1. One-Step Models

Although the one-step model is too simple to encapsulate the essence of AD, it is useful for the understanding of some basic facts concerning optimization of biogas in bioreactors. Consider a one-step model of the form:

$$kS \xrightarrow{r} X + k_1 \mathrm{CH}_4 \tag{1}$$

where one substrate $S$ is consumed by one micro-organism $X$ and produces biogas with reaction rate $r = \mu(S)X$, where $\mu$ is the growth function and $k$ and $k_1$ are pseudo-stoichiometric coefficients. Let $D$ be the dilution rate and $S^{in}$ the concentrations of input substrate. The dynamical equations of the model are [22–24,46,47]

$$
\begin{array}{rcl}
\dot{S} & = & D(S^{in} - S) - k\mu(S)X \\
\dot{X} & = & (\mu(S) - D_1)X
\end{array}
\tag{2}
$$

where $D_1$, the removal rate of the micro-organisms, takes the form

$$
D_1 = \alpha D + a,
\tag{3}
$$

where $a$ is the decay term corresponding to maintenance effects and $\alpha \in (0, 1]$ is a parameter allowing us to decouple the Hydraulic Retention Time, HRT $= 1/D$ and the Solid Retention Time SRT $= 1/(\alpha D)$. The stoichiometric coefficient $k_1$ in (1) appears in the mathematical equations of the model when we consider the biogas flow rate; see Section 2.1.2. The stoichiometric coefficient $k$ can be reduced to 1; see Appendix A.1. However, since the stoichiometric coefficient has its own importance for the biologist, and since our aim is to give the biologist a useful tool for the best operating conditions of the chemostat model, we do not make this reduction and we present the results in the original model (2). The mathematical analysis of (2) is well-known [22,24]. For the convenience of the reader, we recall in this paper the main results and state them using the OD; see Appendix A.2.

### 2.1.1. Steady States

We assume that $\mu$ is not necessarily monotonic, i.e., that the inhibition by substrate $S$ can be taken into account in the model. We make now the following hypothesis.

**Hypothesis 1.** *The function $\mu$ is $\mathcal{C}^1$ and satisfies $\mu(0) = 0$, and there exists $S^m \in (0, +\infty]$, such that $\mu'(S) > 0$ for $0 < S < S^m$. If $S^m < +\infty$, then, in addition, $\mu'(S) < 0$ for $S > S^m$.*

The case $S^m = +\infty$ corresponds to an increasing function. This case is called the *Monod case*, since it is satisfied by the usual Monod growth function

$$
\mu(S) = \frac{mS}{K+S}.
\tag{4}
$$

The case $S^m < +\infty$ corresponds to an increasing and then decreasing function and models the inhibition by the substrate at high concentrations. This case is called the *Haldane case*, since it is satisfied by the usual Haldane growth function

$$
\mu(S) = \frac{mS}{K+S+S^2/K_i}.
\tag{5}
$$

We need to define the break-even concentrations:

**Definition 1.** *When $S^m = +\infty$, the break-even concentration $\lambda(D)$ is the unique solution of equation $\mu(S) = D$. It is defined for $D < \mu(+\infty)$. When $S^m < +\infty$, there can be two break-even concentrations $\lambda(D)$ and $\bar{\lambda}(D)$. They are the solutions of equation $\mu(S) = D$, such that $\lambda(D) < S^m < \bar{\lambda}(D)$. The first one is defined for $0 < D < \mu(S^m)$. The second one is defined for $\mu(+\infty) < D < \mu(S^m)$. They have the same limit value $\lambda(D^m) = \bar{\lambda}(D^m)$ for $D^m = \mu(S^m)$. If $D > \mu(S^m)$, by convention we let $\lambda(D) = +\infty$ and $\bar{\lambda}(D) = +\infty$.*

Besides the washout steady state $F_0 = (S^{in}, 0)$, (2) has the positive steady states

$$
F_1 = \left(\lambda(D_1), \frac{D}{kD_1}(S^{in} - \lambda(D_1))\right), \qquad F_2 = \left(\bar{\lambda}(D_1), \frac{D}{kD_1}(S^{in} - \bar{\lambda}(D_1))\right).
\tag{6}
$$

When $S^m = +\infty$, only $F_1$ exists. The conditions of existence an stability of the steady state, together with the OD of (2), are given in Appendix A.2. Note that $F_1$ is stable whenever its exists, while $F_2$ is unstable whenever its exists.

2.1.2. Steady State Optimization of Biogas Production

The biogas is simply a product of the biological reactions and it has no feedback on the dynamical Equation (2). The biogas flow rate, denoted by $G_{CH_4}$, is proportional to the microbial activity, as proposed in [46,48–50]:

$$G_{CH_4} = k_1 \mu(S^*) X^* \tag{7}$$

where $(S^*, X^*)$ is a steady state of (2). Let us denote by $G_i$, the rate of production of biogas, defined by (7), and evaluated at steady state $F_i$, $i = 0, 1, 2$. One has $G_0 = 0$, and using the components of the steady states $F_1$ and $F_2$ given in (6), $G_1$ and $G_2$ are given by

$$\begin{aligned} G_1(D, S^{in}) &= \tfrac{k_1}{k} D\big(S^{in} - \lambda(\alpha D + a)\big) \quad \text{for} \quad S^{in} \geq \lambda(\alpha D + a), \\ G_2(D, S^{in}) &= \tfrac{k_1}{k} D\big(S^{in} - \bar{\lambda}(\alpha D + a)\big) \quad \text{for} \quad S^{in} \geq \bar{\lambda}(\alpha D + a). \end{aligned} \tag{8}$$

Our aim is to determine the set of operating conditions for which the biogas production is maximal. We consider the biogas flow rate $G_2$ corresponding to the unstable equilibrium $F_2$ because we do not know if this flow rate is always lower than that of the stable equilibrium $F_1$. If it was possible that, for some operating condition $D$ and $S^{in}$, $G_2(D, S^{in}) > G_1(D, S^{in})$, then the problem of the stabilization of the reactor at its unstable steady state $F_2$ by using some feedback control would have been an interesting challenge. However, this possibility is excluded, as stated in the following remark.

**Remark 1.** *Note that $G_2$ is defined if and only if $S^{in} \geq \bar{\lambda}(\alpha D + a)$. Since $\bar{\lambda}(\alpha D + a) > \lambda(\alpha D + a)$, $G_1$ is also defined and we have $G_1(D, S^{in}) > G_2(D, S^{in})$.*

Hence, the operating conditions $D$ and $S^{in}$ which produce the maximum of biogas are obtained by the maximization of $G_1(D, S^{in})$.

**Problem 1.** *Determine the set of operating conditions for which $G_1$ is maximal.*

2.1.3. Steady State Optimization of Biomass Production

AD is used because it allows material to be degraded without producing too much biomass, which is a good thing because in the environmental field we do not really know what to do with the sludge produced. If we want to produce biomass, it is rather in biotechnologies such as pharmaceuticals or food processing that we should be looking. Let us forget about AD for a moment and assume that the industrial goal of the process is the production of micro-organisms. When a continuous culture system is viewed as a production process, its performance may be judged by the quantity of bacteria produced, which is called the productivity of biomass. The total output from a continuous culture unit in the steady state is equal to the product of flow rate and concentration of organisms. Therefore, the productivity of (2) at steady state $(S^*, X^*)$ is given by [20,47]

$$P = Q X^* \tag{9}$$

where $Q = VD$ is the flow rate, and $V$ is the volume of the CSTR. Let us denote by $P_i$, the productivity evaluated at steady state $F_i$, $i = 0, 1, 2$. One has $P_0 = 0$ and using the components of the steady states $F_1$ and $F_2$, given in (6), $P_1$ and $P_2$ are given by

$$\begin{aligned} P_1(D, S^{in}) &= \tfrac{VD^2}{k(\alpha D + a)}\big(S^{in} - \lambda(\alpha D + a)\big) \quad \text{for} \quad S^{in} \geq \lambda(\alpha D + a), \\ P_2(D, S^{in}) &= \tfrac{VD^2}{k(\alpha D + a)}\big(S^{in} - \bar{\lambda}(\alpha D + a)\big) \quad \text{for} \quad S^{in} \geq \bar{\lambda}(\alpha D + a). \end{aligned} \tag{10}$$

Our aim is to determine the set of operating conditions for which the productivity is maximal. Note that, as for the biogas flow rate, the productivity at $F_1$ is greater the the productivity at $F_2$: $P_1(D, S^{in}) > P_2(D, S^{in})$. Hence, the operating conditions $D$ and $S^{in}$ that maximize productivity are obtained by maximizing $P_1(D, S^{in})$.

**Problem 2.** *Determine the set of operating conditions for which $P_1$ is maximal.*

### 2.1.4. The Case without Mortality

Note that when $a = 0$, we have

$$G_1(D, S^{in}) = \tfrac{k_1}{k} D(S^{in} - \lambda(\alpha D)), \qquad G_2(D, S^{in}) = \tfrac{k_1}{k} D(S^{in} - \bar{\lambda}(\alpha D)),$$
$$P_1(D, S^{in}) = \tfrac{V}{k\alpha} D(S^{in} - \lambda(\alpha D)), \qquad P_2(D, S^{in}) = \tfrac{V}{k\alpha} D(S^{in} - \bar{\lambda}(\alpha D)).$$

Therefore, $G_i$ and $P_i$, $i = 1, 2$ are proportional. Hence, we can make the following remark.

**Remark 2.** *When $a = 0$, optimizing $P_1$, given by (10), is the same as optimizing $G_1$, given by (8); that is, Problems 1 and 2 have the same solution. However, this is no longer true when $a > 0$.*

For increasing functions (i.e., $S^m = +\infty$), in the case $a = 0$, the equivalent Problems 1 and 2 have been solved in [51]; in the case $a > 0$, Problem 1 has been solved in [52] and Problem 2 in [53]. In Sections 3.1.1 and 3.1.5, we will give the solutions to these problems in the more general case where the growth function $\mu$ satisfies the Hypothesis 1 and is not necessarily monotonic.

### 2.2. Two-Step Models

We consider the general two-step model with a cascade of two biological reactions, where one substrate $S_1$ is consumed by one microorganism $X_1$ (*acidogenic* bacteria, in the AM2 model), to produce a product $S_2$ that serves as the main limiting substrate for a second microorganism $X_2$ (*methanogenic* bacteria in the AM2 model) as schematically represented by the following reaction scheme (see [25]):

$$k_1 S_1 \xrightarrow{r_1} X_1 + k_2 S_2, \quad k_3 S_2 \xrightarrow{r_2} X_2 + k_4 CH_4 \tag{11}$$

where $r_1 = \mu_1(S_1)X_1$ and $r_2 = \mu_2(S_2)X_2$ are the kinetics of the reactions and $k_i$, $i = 1, \ldots, 4$ are pseudo-stoichiometric coefficients. In fact, biological reactions also produce $CO_2$; see Equations (1) and (2) in [25]. However, since in this section we are only interested in the biogas production, we do not focus on the $CO_2$ production. Let $D$ be the dilution rate and $S_1^{in}$ and $S_2^{in}$ the concentrations of input substrates $S_1$ and $S_2$, respectively. The dynamical equations of the model take the form:

$$\begin{aligned}
\dot{S}_1 &= D(S_1^{in} - S_1) - k_1 \mu_1(S_1)X_1, \\
\dot{X}_1 &= (\mu_1(S_1) - D_1)X_1, \\
\dot{S}_2 &= D(S_2^{in} - S_2) + k_2 \mu_1(S_1)X_1 - k_3 \mu_2(S_2)X_2, \\
\dot{X}_2 &= (\mu_2(S_2) - D_2)X_2,
\end{aligned} \tag{12}$$

where, as in (3), the removal rates of the micro-organisms $D_1$ and $D_2$ take the form

$$D_i = \alpha_i D + a_i, \quad i = 1, 2, \tag{13}$$

where $\alpha_i \in (0, 1]$, $i = 1, 2$, is a parameter allowing us to decouple the HRT and the SRT. This decoupling is necessary when considering technology such as systems where biomass is fixed onto supports (as in fixed or fluidized bed reactors) or still retained in the system by membranes such as in MBRs (Membrane Bioreactors); see [54,55]. The model (12) is an

extension of the AM2 model presented in [25], with $\alpha_1 = \alpha_2$, $a_1 = a_2 = 0$, and kinetics $\mu_1$ and $\mu_2$ of Monod and Haldane types, respectively.

The pseudo-stoichiometric coefficients $k_i$ in (12) can be reduced to 1; see Appendix B.1. However, since these coefficients have their own importance for the biologist and since our aim is to discuss the best operating conditions, we do not make this reduction and we present the results in the original model (12). The model has a cascade structure which renders its analysis easy. We give in Appendix B.3 the main results on the existence and stability of the steady states of (12), and we express them using the OD.

### 2.2.1. Steady States

We consider (12) with general kinetics functions $\mu_1$ and $\mu_2$, satisfying the following qualitative properties:

**Hypothesis 2.** *The function $\mu_1$ is $\mathcal{C}^1$, $\mu_1(0) = 0$, $\mu_1'(S_1) > 0$ for $S_1 > 0$. Let $m_1 = \mu_1(+\infty)$.*

**Hypothesis 3.** *The function $\mu_2$ is $\mathcal{C}^1$, $\mu_2(0) = 0$, $\mu_2(+\infty) = 0$, and there exists $S_2^m > 0$ such that $\mu_2'(S_2) > 0$ for $0 < S_2 < S_2^m$, and $\mu_2'(S_2) < 0$ for $S_2 > S_2^m$.*

We consider the break-even concentrations as stated in Definition 1. The growth function $\mu_1$ admits only one break-even concentration, denoted $\lambda_1$, while the growth function $\mu_2$ admits two break-even concentrations, which will be denoted $\lambda_2$ and $\bar{\lambda}_2$. We summarize in Table 1 the definitions of these break-even concentrations, together with two auxiliary functions that are used in the description of the biogas flow-rates at steady states of (12).

**Table 1.** Break-even concentrations and auxiliary functions.

| |
| --- |
| $\lambda_1(D)$ is the unique solution of equation $\mu_1(S_1) = D$, for $D < m_1$ |
| $\lambda_2(D) < \bar{\lambda}_2(D)$ are the solutions of equation $\mu_2(S_2) = D$, for $D < \mu_2(S_2^m)$<br>$\lambda(0) = 0$, $\bar{\lambda}_2(0) = +\infty$ and $\lambda(D) = \bar{\lambda}_2(D)$ for $D = \mu_2(S_2^m)$<br>$H_1(D) = \lambda_2(D_2) + \frac{k_2}{k_1}\lambda_1(D_1)$,<br>$H_2(D) = \bar{\lambda}_2(D_2) + \frac{k_2}{k_1}\lambda_1(D_1)$ |

The system (12) can have up to six steady states, denoted $E_{ij}$, where $i = 0, 1$ and $j = 0, 1, 2$. The components of the steady states are given in Table A3. The existence and stability conditions of the steady states of (12) are given in Appendix B.3. Note that $E_{11}$ is stable whenever it exists, while $E_{01}$ is stable if and only if it exists and $E_{11}$ does not exist. Moreover the steady states $E_{02}$ and $E_{12}$ are unstable whenever they exist.

### 2.2.2. Steady State Optimization of Biogas Production

As in the one-step model, the biogas is simply a product of the biological reactions and it has no feedback on the dynamical Equation (12). As we noticed in (7), the mass flow of the methane production, denoted by $G_{CH_4}$, is proportional to the microbial activity (see Equation (12) in [25]):

$$G_{CH_4} = k_4 \mu_2(S_2) X_2.$$

Let us denote by $G_{ij}$ the production of biogas at steady states $E_{ij}$ for $i = 0, 1$ and $j = 0, 1, 2$. Using the components of the steady states given in Table A3, it is seen that $G_{00} = G_{10} = 0$ and $G_{ij}$ for $i = 0, 1$ and $j = 1, 2$ are defined as in Table 2.

**Table 2.** The biogas production at steady state $E_{ij}$, $i = 0, 1$, $j = 1, 2$; $\lambda_2(D)$, $\bar{\lambda}_2(D)$ and $H_j(D)$, $j = 1, 2$, are defined in Table 1.

| Biogas Production | Domain of Definition |
|---|---|
| $G_{01}\left(D, S_2^{in}\right) = \frac{k_4}{k_3} D\left(S_2^{in} - \lambda_2(D_2)\right)$ | $\lambda_2(D_2) \leq S_2^{in}$ |
| $G_{02}\left(D, S_2^{in}\right) = \frac{k_4}{k_3} D\left(S_2^{in} - \bar{\lambda}_2(D_2)\right)$ | $\bar{\lambda}_2(D_2) \leq S_2^{in}$ |
| $G_{11}\left(D, S_1^{in}, S_2^{in}\right) = \frac{k_4}{k_3} D\left(S_2^{in} + \frac{k_2}{k_1} S_1^{in} - H_1(D)\right)$ | $\lambda_1(D_1) \leq S_1^{in}, H_1(D) \leq S_2^{in} + \frac{k_2}{k_1} S_1^{in}$ |
| $G_{12}\left(D, S_1^{in}, S_2^{in}\right) = \frac{k_4}{k_3} D\left(S_2^{in} + \frac{k_2}{k_1} S_1^{in} - H_2(D)\right)$ | $\lambda_1(D_1) \leq S_1^{in}, H_2(D) \leq S_2^{in} + \frac{k_2}{k_1} S_1^{in}$ |

Our aim is to find set of operating conditions for which the flow rate of biogas is maximal.

**Remark 3.** *We always have* $G_{01} > G_{02}$ *and* $G_{11} > G_{12}$; *see Section 3.2.1.*

Hence, the operating conditions $D$, $S_1^{in}$, and $S_2^{in}$, which produce the maximum of biogas, are obtained by the maximization of $G_{01}(D, S_2^{in})$ or $G_{11}(D, S_1^{in}, S_2^{in})$. The main problem is then to compare the maximum of biogas production $G_{11}$ at $E_{11}$, where both species are present, with the maximum of biogas production $G_{01}$ at $E_{01}$ where species $X_1$ is extinct and species $X_2$ is present. Surprisingly, the optimal biogas production does not always occur at $E_{11}$, as was noticed by [11,21,31]. Therefore we have to solve the following problem.

**Problem 3.** *Determine the sets of operating conditions, for which* $G_{01}$ *and* $G_{11}$ *are maximal. Compare the maximum of* $G_{01}$ *to that of* $G_{11}$.

## 3. Results

*3.1. One-Step Models*

The OD of the one-step model (2) is described in Appendix A.2.

### 3.1.1. Best Operating Conditions for Biogas Production

Let $G_1$ defined by (8) and $S^{in}$ fixed. Our aim is to maximize the function $D \mapsto G_1(D, S^{in})$. Note that this function is proportional to the function $G$ defined by

$$G(D) = D(S^{in} - \lambda(\alpha D + a)). \tag{14}$$

The function $G$ is depending on the parameter $S^{in}$. It is defined for $D \in I(S^{in})$, where the interval $I(S^{in})$ is given by

$$I(S^{in}) = \begin{cases} [0, \delta(S^{in})] & \text{if } S^{in} < S^m \\ [0, \delta(S^m)] & \text{if } S^{in} \geq S^m \end{cases} \quad \text{with} \quad \delta(S) = \frac{\mu(S) - a}{\alpha} \tag{15}$$

The function $G_1$ has an absolute maximum if $G$ has one and this maximum is reached at the same point where $G$ reaches its maximum. By the *Extreme Value Theorem*, since $G$ is continuous on the closed interval $I(S^{in})$, it must attain a maximum. Let us consider the set of arguments of the maximum of $G$, denoted by $g(S^{in})$ and defined by

$$g(S^{in}) = \underset{D \in I(S^{in})}{\text{argmax}} \, G := \left\{ D^* \in I(S^{in}) : G(D) \leq G(D^*) \text{ for all } D \in I(S^{in}) \right\}. \tag{16}$$

To obtain the maximum value of $G(D)$, we differentiate (14) with respect to $D$, and we solve the equation $G'(D) = 0$. The derivative of $G$ is given by

$$G'(D) = S^{in} - \gamma(D)$$

where $\gamma$ is defined by

$$\gamma(D) = \lambda(\alpha D + a) + \alpha D \lambda'(\alpha D + a). \tag{17}$$

**Remark 4.** *Since $\mu(\lambda(D)) = D$, we have $\lambda'(D) = 1/\mu'(\lambda(D))$. Therefore the function $\gamma$ is written*

$$\gamma(D) = \lambda(\alpha D + a) + \frac{\alpha D}{\mu'(\lambda(\alpha D + a))}.$$

We have the following result

**Proposition 1.** *Let $D^* \in g(S^{in})$. We have $S^{in} = \gamma(D^*)$, where $\gamma$ is defined by (17).*

**Proof.** The proof is given in Appendix A.3.1.  $\square$

Therefore, the curve

$$\Gamma = \left\{ (D, S^{in}) : S^{in} = \gamma(D) \right\} \tag{18}$$

of SOP contains the operating conditions for which $G_1$ is maximal.

In Figure 1, we plot the $\Gamma$ curve in the OD of (2). We have shown a curve $\Gamma$, which is the graph of an increasing function. However, this does not always happen; see Appendix A.5.5. When $\Gamma$ is not increasing, there may be several maxima of the biogas flow. In Section 3.1.3, we give sufficient conditions for the maximum to be unique. Since $\lambda'(D) > 0$, we deduce that $\gamma(D) > \lambda(\alpha D + a)$ for $D > 0$. On the other hand,

$$\gamma(0) = \lambda(a), \quad \text{and} \quad \lim_{D \to \delta(S^m)} \gamma(D) = +\infty.$$

From these properties we deduce the following remark.

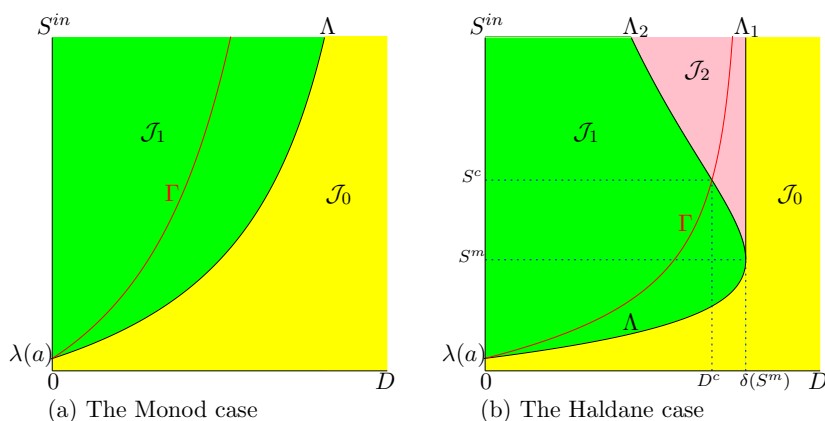

(a) The Monod case          (b) The Haldane case

**Figure 1.** The OD of (2). The curve $\Gamma$ is the set of best operating conditions.

**Remark 5.** *If $S^m = +\infty$, the curve $\Gamma$ is contained in the region $\mathcal{J}_1$ (the green region) of the OD, see Figure 1a. If $S^m < +\infty$, $\Gamma$ is contained in $\mathcal{J}_1 \cup \mathcal{J}_2$ (the green and pink regions), and, since $\mu'(S^m) = 0$, the vertical line $\Lambda_1$ is an asymptote of $\Gamma$, see Figure 1b. Note that in the Haldane case $(S^m < \infty)$, the curve $\Gamma$ enters in the bistability region $\mathcal{J}_2$ at point $(D^c, S^c)$.*

3.1.2. How to Determine the Maximum of Biogas Production

From Proposition 1, to obtain $g(S^{in})$, we must solve the equation $S^{in} = \gamma(D)$. However, this equation can be complicated to solve because $\gamma(D)$ is itself defined by $\lambda(D)$, which is the solution of the equation $\mu(S) = D$. We have at our disposal another description of $g(S^{in})$. Indeed, we can write

$$G(D) = \tfrac{1}{\alpha} H(\lambda(\alpha D + a)), \tag{19}$$

where $H$ is defined by

$$H(S) = (\mu(S) - a)(S^{in} - S), \quad \text{for} \quad \lambda(a) \le S \le S^{in} \tag{20}$$

From (19), it is deduced that the absolute maximum of $G$ corresponds to the absolute maximum of $H$ and vice versa. To obtain the maximum value of $H(S)$, we differentiate $H$ with respect to $S$ and we solve the equation $H'(S) = 0$. The derivative of $H$ is given by

$$H'(S) = \mu'(S)(S^{in} - S) - \mu(S) + a.$$

Hence, $H'(S) = 0$ if and only if $S^{in} = \eta(S)$, where $\eta(S)$ is defined by

$$\eta(S) = S + \frac{\mu(S) - a}{\mu'(S)} \quad \text{for} \quad S \ge \lambda(a). \tag{21}$$

We have the following result.

**Proposition 2.** *Let $S^*$ be the maximum of $H$ on $(\lambda(a), S^{in})$. Let $D^* = \frac{\mu(S^*) - a}{\alpha}$. Then $D^* \in g(S^{in})$. Moreover, we have $S^{in} = \eta(S^*)$, where $\eta$ is defined by (21).*

**Proof.** The proof is given in Appendix A.3.2. □

**Remark 6.** *With the first method, we must first solve the equation $\mu(S) = D$ to obtain $\lambda(D)$ and then solve the equation $\gamma(D) = S^{in}$ to obtain the optimal $D^* \in g(S^{in})$. With the second method, we simply solve the equation $\eta(S) = S^{in}$ to get the maximum $S^*$ and then take $D^* = \frac{\mu(S^*) - a}{\alpha} \in g(S^{in})$.*

3.1.3. Uniqueness of the Maximum

Hypothesis 1 is not enough to guarantee that the biogas flow rate admits a unique global maximum; see Appendix A.5.5. We make the following hypothesis.

**Hypothesis 4.** *For all $S^{in} > 0$, $g(S^{in})$, defined by (16), has a unique element, which is denoted by $D_G^*(S^{in})$.*

From Proposition 1 we deduce then the answer to Problem 1: assume that Hypotheses 1 and 4 are satisfied. Then, the set of best operating conditions for biogas production of (2) is the curve $\Gamma$ of SOP defined by:

$$\Gamma = \left\{ (D, S^{in}) : S^{in} = \gamma(D) \right\} = \left\{ (D, S^{in}) : D = D_G^*(S^{in}) \right\}. \tag{22}$$

From Propositions 1 and 2, it is deduced that Hypothesis 4 is satisfied when the equations

$$S^{in} = \gamma(D) \quad \text{or} \quad S^{in} = \eta(S)$$

have a unique solution. A sufficient condition for this is that the functions $\gamma(D)$ and $\eta(S)$ are increasing. The following result gives sufficient conditions for Hypothesis 4 to be valid.

**Lemma 1.** *Assume that Hypothesis 1 is satisfied and, in addition, $\mu$ is $C^2$. The following conditions are equivalent*

1. $\gamma' > 0$ on $\left(0, \frac{\mu(S^m) - a}{\alpha}\right)$.
2. $(\mu - a)\mu'' < 2(\mu')^2$ on $(\lambda(a), S^m)$.
3. $\left(\frac{1}{\mu - a}\right)'' > 0$ on $(\lambda(a), S^m)$.
4. $\eta' > 0$ on $(\lambda(a), S^m)$.

*If these equivalent conditions are satisfied, then Hypothesis 4 is satisfied. If $\mu'' < 0$ on $(\lambda(a), S^m)$, then the conditions are satisfied.*

**Proof.** The proof is given in Appendix A.3.3. $\square$

### 3.1.4. Best Operating Conditions

We first analyse the Monod case ($S^m = \infty$). We show in Figure 2 the set $\Gamma$ of best operating conditions and we describe how to use this set to obtain practically the maximum of biogas production. Let $S^{in}$ be fixed. The intersections of $\Gamma$ and $\Lambda$ with the horizontal line where $S^{in}$ is kept constant define the values $D_G^*(S^{in})$, defined in Hypothesis 4, and $\delta(S^{in}) = \frac{\mu(S^{in}) - a}{\alpha}$, defined by (15), see Figure 2a. The function $D \mapsto G_1(D, S^{in})$ is defined on $[0, \delta(S^{in})]$ and attains its maximum $G^*(S^{in})$ for $D = D_G^*(S^{in})$; see Figure 2b.

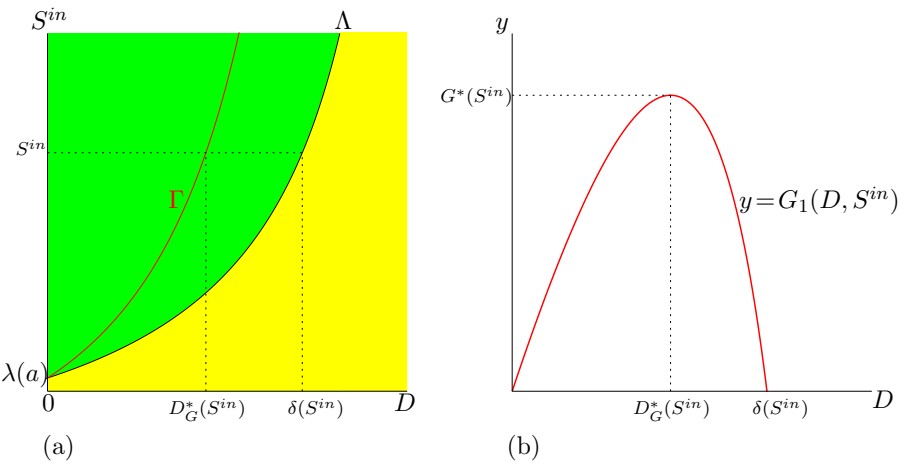

(a)
(b)

**Figure 2.** The best operating conditions of biogas flow rate for the Monod case. (**a**): The curve $\Gamma$ in SOP shows the optimal value $D_G^*(S^{in})$. (**b**): The function $D \mapsto G_1(D, S^{in})$ is defined on $[0, \delta(S^{in})]$, and attains its maximum, $G^*(S^{in})$, for $D = D_G^*(S^{in})$.

In the Haldane case ($S^m < \infty$), the description is a little more complicated. If $S^{in}$ is fixed, the function $D \mapsto G_1(D, S^{in})$ attains its maximum $G_1^*(S^{in})$ for $D = D_G^*(S^{in})$, obtained by taking the intersection of $\Gamma$ with the horizontal line where $S^{in}$ is kept constant, as it is seen in Figure 3. However, there exist two threshold values $S^c$ and $S^m$, depicted in Figure 1b. If $S^{in} \leq S^m$, only $G_1$ is defined (see Figure 3a) while $G_1$ and $G_2$ are both defined when $S^{in} > S^m$ (see Figure 3b,c). On the other hand, if $S^{in} > S^c$, then the dilution rate $D_G^*(S^{in})$, which maximises biogas production, corresponds to the bistability mode of the chemostat; see Figure 3c. More precisely, we make the following remark.

**Remark 7.** *Assume that Hypotheses 1 and 4 hold. Let $D = D^c$ be the unique solution to equation $\gamma(D) = \bar{\lambda}(\alpha D + a)$. Let $S^c = \gamma(D^c)$.*

- *If $S^{in} < S^c$ then for the operating parameters $S^{in}$ and $D = D_G^*(S^{in})$, $F_1$ is GAS.*
- *If $S^{in} > S^c$ then for the operating parameters $S^{in}$ and $D = D_G^*(S^{in})$, $F_0$ and $F_1$ are both stable.*

Indeed, since $\gamma$ is increasing and $\bar{\lambda}$ is decreasing, curves $\Gamma$ and $\Lambda_2$ have a unique intersection point $(D^c, S^c)$; see Figure 1b. The OD shows that if $S^{in} < S^c$ then $(D_G^*(S^{in}), S^{in}) \in \mathcal{J}_1$, that is to say, the best operating conditions are in the green region $\mathcal{J}_1$, where $F_1$ is GAS and if $S^{in} > S^c$, then $(D_G^*(S^{in}), S^{in}) \in \mathcal{J}_2$; that is to say, the best operating conditions are in the pink region $\mathcal{J}_2$ of bistability of $F_0$ and $F_1$.

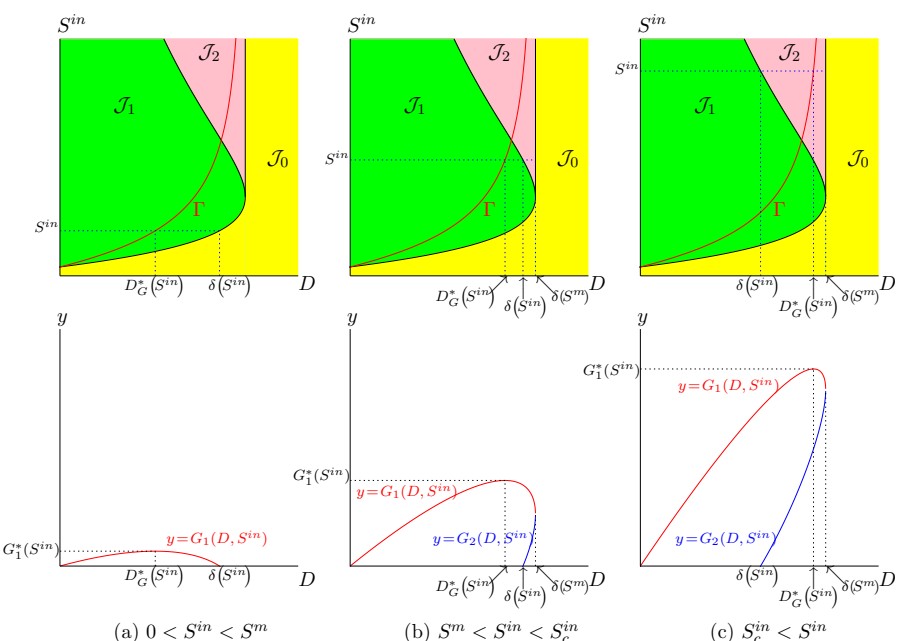

**Figure 3.** The set of best operating conditions $\Gamma$ (in red) shows the optimal dilution rate $D_G^*(S^{in})$ corresponding to three typical values of $S^{in}$.

Figure 3 shows three typical values of $S^{in}$ and the corresponding optimal dilution rates $D_G^*(S^{in})$. The corresponding biogas productions are depicted in the same figure. The main results are summarized as follows:

- If $S^{in} < S^m$, the biogas production $G_1(D, S^{in})$ is defined for $D \in [0, \delta(S^{in})]$; see Figure 3a.
- If $S^{in} > S^m$, the biogas production $G_1(D, S^{in})$ is defined for $D \in [0, \delta(S^m)]$, and the biogas production $G_2(D, S^{in})$ is defined for $D \in [\delta(S^{in}), \delta(S^m)]$; see Figure 3b,c.
- If $S^{in} < S^c$, and the chemostat is operated at the optimal dilution rate $D_G^*(S^{in})$, then the system converges towards the positive steady state $F_1$ giving the maximum of biogas; see Figure 3a,b.
- If $S^{in} > S^c$ and the chemostat is operated at the optimal dilution rate $D_G^*(S^{in})$, then, according to the initial condition, the system converges either to the positive steady state $F_1$, giving maximum biogas, or the washout steady state $F_0$, with no biogas production; see Figure 3c.

### 3.1.5. Best Operating Conditions for Biomass Production

Let $P_1$ be defined by (10) and $S^{in}$ fixed. Our aim is to maximise the function $D \mapsto P_1(D, S^{in})$. Note that this function is proportional to the function $P : D \mapsto p(D)$ defined by

$$P(D) = \frac{D^2}{\alpha D + a}\left(S^{in} - \lambda(\alpha D + a)\right), \quad \text{for} \quad D \in I(S^{in}) \tag{23}$$

where $I(S^{in})$ is defined by (15). Therefore $P_1$ has an absolute maximum if $P$ has one and this maximum is reached at the same point where $P$ reaches its maximum. As in the case of the biogas flow rate, we consider the arguments of the maximum of $P$

$$p(S^{in}) = \underset{D \in I(S^{in})}{\text{argmax}}\, p := \left\{ D^* \in I(S^{in}) : P(D) \leq P(D^*) \text{ for all } D \in I(S^{in}) \right\}. \tag{24}$$

To obtain the maximum value of $P(D)$, we differentiate (23) with respect to $D$, and we solve the equation $P'(D) = 0$. The derivative of $P$ is given by

$$P'(D) = \frac{D(\alpha D + 2a)}{(\alpha D + a)^2}\left(S^{in} - \pi(D)\right)$$

where $\pi$ is defined by

$$\pi(D) = \lambda(\alpha D + a) + \frac{\alpha D(\alpha D + a)}{\alpha D + 2a}\lambda'(\alpha D + a). \tag{25}$$

**Remark 8.** *Using* $\lambda'(D) = 1/\mu'(\lambda(D))$*, the function* $\pi$ *can be written*

$$\pi(D) = \lambda(\alpha D + a) + \frac{\alpha D(\alpha D + a)}{(\alpha D + 2a)\mu'(\lambda(\alpha D + a))}.$$

We have the following result

**Proposition 3.** *Let* $D^* \in p(S^{in})$*. We have* $S^{in} = \pi(D^*)$*, where* $\pi$ *is defined by* (25).

**Proof.** The proof is given in Appendix A.4.1. □

Therefore, the curve

$$\Pi = \left\{ (D, S^{in}) : S^{in} = \pi(D) \right\} \tag{26}$$

of SOP contains the operating conditions for which $P_1$ is maximal. In Figure 4, this set is shown in the OD depicted in Figure 1, together with the set $\Gamma$. Note that if $a > 0$, then

$$\lambda(\alpha D + a) < \pi(D) < \gamma(D). \tag{27}$$

Therefore, curve $\Pi$ is above curve $\Lambda$ and below curve $\Gamma$; see Figure 4.

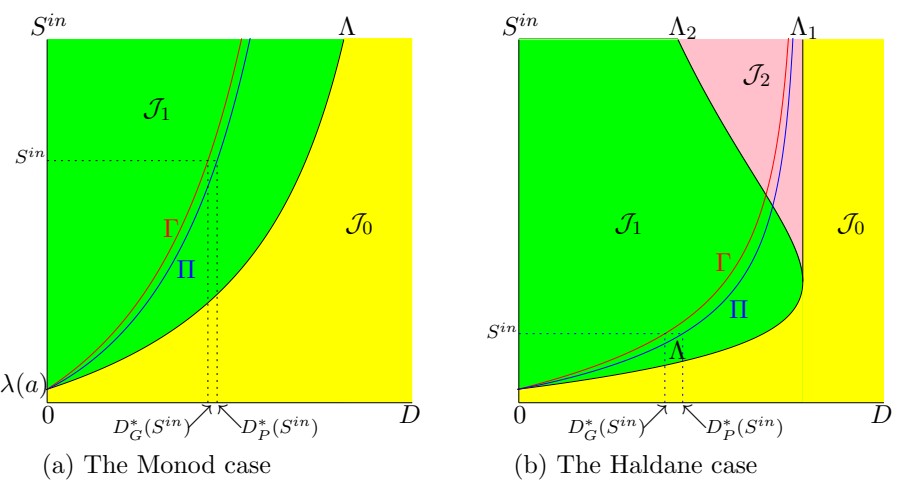

(a) The Monod case      (b) The Haldane case

**Figure 4.** The curves $\Gamma$ (in red) and $\Pi$ (in blue).

3.1.6. How to Determine the Maximum of Biomass Production?

From Proposition 3, to obtain $p(S^{in})$ we must solve the equation $S^{in} = \pi(D)$, which can be difficult to solve. We have at our disposal another description of $p(S^{in})$. We can write

$$P(D) = \frac{1}{\alpha^2}Q(\lambda(\alpha D + a)), \tag{28}$$

where $Q$ is defined by

$$Q(S) = \frac{(\mu(S) - a)^2}{\mu(S)}(S^{in} - S), \quad \text{for} \quad \lambda(a) \le S \le S^{in} \tag{29}$$

From (28), it is deduced that the absolute maximum of $P$ corresponds to the absolute maximum of $Q$ and vice versa. To obtain the maximum value of $Q(S)$, we differentiate $Q$ with respect to $S$, and we solve the equation $Q'(S) = 0$. The derivative of $Q$ is given by

$$Q'(S) = \frac{(\mu(S) - a)(\mu(S) + a)\mu'(S)}{(\mu(S))^2}(S^{in} - S) - \frac{(\mu(S) - a)^2}{\mu(S)}.$$

Hence, $Q'(S) = 0$ if and only if $S^{in} = \rho(S)$, where $\rho(S)$ is defined by

$$\rho(S) = S + \frac{(\mu(S)-a)\mu(S)}{(\mu(S)+a)\mu'(S)}, \quad \text{for} \quad S \geq \lambda(a). \tag{30}$$

More precisely, we have the following result.

**Proposition 4.** *Let $S^*$ be the maximum of $p$ on $(\lambda(a), S^{in})$. Then $D^* = \frac{\mu(S^*)-a}{\alpha}$. We have $D^* \in p(S^{in})$. Moreover, we have $S^{in} = \rho(S^*)$, where $\rho$ is defined by (30).*

**Proof.** The proof is given in Appendix A.4.2. $\square$

**Remark 9.** *With the first method we must first solve the equation $\mu(S) = D$ to obtain $\lambda(D)$, and then solve the equation $\pi(D) = S^{in}$ to obtain the optimal $D^* \in p(S^{in})$. With the second method, we simply solve the equation $\rho(S) = S^{in}$ to get the maximum $S^*$ and then take $D^* = \frac{\mu(S^*)-a}{\alpha} \in p(S^{in})$.*

3.1.7. Uniqueness of the Maximum

Hypothesis 1 is not enough to guarantee that the biomass productivity admits a unique global maximum; see Appendix A.5.5. We make the following hypothesis.

**Hypothesis 5.** *For all $S^{in} > 0$, $p(S^{in})$, defined by (24), has a unique element, which is denoted by $D_P^*(S^{in})$.*

From Proposition 3, we obtain the answer to Problem 2: Assume that Hypotheses 1 and 5 hold. Then, the set of best operating conditions for the productivity of (2) is the curve $\Pi$ of SOP defined by:

$$\Pi = \left\{ (D, S^{in}) : S^{in} = \pi(D) \right\} = \left\{ (D, S^{in}) : D = D_P^*(S^{in}) \right\}. \tag{31}$$

From (27), we deduce that if $a > 0$, then $D_G^*(S^{in}) < D_P^*(S^{in})$; see Figure 4.

From Propositions 3 and 4, it is deduced that the uniqueness of $D_P^*(S^{in})$ is guaranteed when the equations

$$S^{in} = \pi(D) \quad \text{or} \quad S^{in} = \rho(S)$$

have a unique solution. A sufficient condition for this is that the functions $\pi(D)$ and $\rho(S)$ are increasing. The following result gives sufficient conditions for Hypothesis 5 to be valid.

**Lemma 2.** *Assume that Hypothesis 1 is satisfied and, in addition, $\mu$ is $C^2$. The following conditions are equivalent*

1. $\pi' > 0$ on $\left(0, \frac{\mu(S^m)-a}{\alpha}\right)$.
2. $\frac{(\mu-a)(\mu+a)}{\mu+2a}\mu'' < 2(\mu')^2$ on $(\lambda(a), S^m)$.
3. $\rho' > 0$ on $(\lambda(a), S^m)$.

*If these equivalent conditions are satisfied, then Hypothesis 5 is satisfied. If $\mu'' < 0$ on $(\lambda(a), S^m)$ or $\left(\frac{1}{\mu-a}\right)'' > 0$ on $(\lambda(a), S^m)$, then the conditions are satisfied.*

**Proof.** The proof is given in Appendix A.4.3. $\square$

3.1.8. The Case without Mortality

The functions $\gamma$, $H$, and $\eta$, defined by (17), (20), and (21), respectively, that were used for the optimization of the biogas flow rate $G$ are summarized in Table 3. Note that the functions $G$ and $H$ are related by formula (19). Similarly, the functions $\pi$, $Q$, and $\rho$, defined by (25), (29) and (30), respectively, which were used for the optimization of the productivity $P$, are summarized in Table 3. Note that the functions $P$ and $Q$ are related by formula (28).

**Table 3.** The functions $\gamma$, $H$ and $\eta$ used for the optimization of the biogas flow rate $G$. The functions $\pi$, $Q$ and $\rho$ used for the optimization of the productivity $P$. Note that $G(D) = \frac{1}{\alpha}H(\lambda(\alpha D + a))$ and $P(D) = \frac{1}{\alpha}Q(\lambda(\alpha D + a))$.

| **Biogas Production** | **Biomass Productivity** |
|---|---|
| $G(D) = D(S^{in} - \lambda(\alpha D + a))$ | $P(D) = \frac{D^2}{\alpha D + a}(S^{in} - \lambda(\alpha D + a))$ |
| $\gamma(D) = \lambda(\alpha D + a) + \alpha D \lambda'(\alpha D + a)$ | $\pi(D) = \lambda(\alpha D + a) + \frac{\alpha D(\alpha D + a)}{\alpha D + 2a}\lambda'(\alpha D + a)$ |
| $H(S) = (\mu(S) - a)(S^{in} - S)$ | $Q(S) = \frac{(\mu(S) - a)^2}{\mu(S)}(S^{in} - S)$ |
| $\eta(S) = S + \frac{\mu(S) - a}{\mu'(S)}$ | $\rho(S) = S + \frac{(\mu(S) - a)\mu(S)}{(\mu(S) + a)\mu'(S)}$ |

| Biogas Production = Biomass Productivity ($a = 0$) | | |
|---|---|---|
| $G(D) = \alpha P(D) = D(S^{in} - \lambda(\alpha D))$ | | |
| $\gamma(D) = \pi(D) = \lambda(\alpha D) + \alpha D \lambda'(\alpha D)$ | | |
| $H(S) = Q(S) = \mu(S)(S^{in} - S)$ | | |
| $\eta(S) = \rho(S) = S + \frac{\mu(S)}{\mu'(S)}$ | | |

Table 3 shows that in the case without mortality, one has $G = \alpha P$, $\gamma = \pi$, $H = Q$, and $\eta = \rho$. Hence, if $a = 0$, we have $D_G^*(S^{in}) = D_P^*(S^{in})$. In the following, this value is referred to as $D^*(S^{in})$. Therefore, for the optimization of the biogas flow rate or the productivity of the biomass, a first method consists in solving the equation

$$\lambda(\alpha D) + \alpha D \lambda'(\alpha D) = S^{in}.$$

to obtain the optimal value of the dilution rate $D^*(S^{in})$. The second method consists in solving the equation

$$\eta(S) = S^{in}, \quad \text{where} \quad \eta(S) := S + \frac{\mu(S)}{\mu'(S)} \tag{32}$$

to get the maximum $S^*(S^{in})$ and then take $D^*(S^{in}) = \frac{1}{\alpha}\mu(S^*(S^{in}))$. Hence, without loss of generality, one can put $\alpha = 1$ and solve Equation (32) or equation

$$\gamma(D) = S^{in}, \quad \text{where} \quad \gamma(D) := \lambda(D) + D\lambda'(D). \tag{33}$$

The results of Lemmas 1 and 2 become the same in the case that $a = 0$. We summarize them below, in this special case.

**Lemma 3.** *Assume that Hypothesis 1 is satisfied and, in addition $\mu$ is $C^2$. The following conditions are equivalent*

1. $\gamma' > 0$ *on* $(0, \mu(S^m))$, *where $\gamma$ is defined in* (33).
2. $\mu\mu'' < 2(\mu')^2$ *on* $(0, S^m)$.
3. $\left(\frac{1}{\mu}\right)'' > 0$ *on* $(0, S^m)$.
4. $\eta' > 0$ *on* $(0, S^m)$, *where $\eta$ is defined in* (32).

*If these equivalent conditions are satisfied, then each of Equations (32) and (33) has a unique solution; i.e., Hypothesis 4 is satisfied. If $\mu'' < 0$ on $(0, S^m)$, then the conditions are satisfied.*

In Appendix A.5, we apply the preceding results to various growth functions that were considered in the literature.

### 3.2. Two-Step Models

The steady state and their stability of the two-step model (12) are given in Appendix B.2, and the OD is described in Appendix B.3.

3.2.1. Comparison of Biogas Flow Rates

Recall that $E_{11}$ is stable whenever it exists; $E_{01}$ can be stable but is unstable whenever $E_{11}$ exists, and $E_{02}$ and $E_{12}$ are unstable whenever they exist. Is it possible that for some operating condition $D$, $S_1^{in}$, and $S_2^{in}$, the biogas production at an unstable steady state is greater than at a stable one? This possibility is excluded, as is stated in the following result.

**Proposition 5.**
- *For all operating conditions $D$ and $S_2^{in}$ where $G_{02}$ is defined, then $G_{01}$ is also defined, and $G_{01}(D, S_2^{in}) > G_{02}(D, S_2^{in})$.*
- *For all operating conditions $D$, $S_1^{in}$ and $S_2^{in}$ where $G_{12}$ is defined, then $G_{11}$ is also defined, and $G_{11}(D, S_1^{in}, S_2^{in}) > G_{12}(D, S_1^{in}, S_2^{in})$.*
- *For all operating conditions $D$, $S_1^{in}$ and $S_2^{in}$ where $G_{01}$ and $G_{11}$ are both defined, we have $G_{11}(D, S_1^{in}, S_2^{in}) > G_{01}(D, S_2^{in})$.*

**Proof.** The proof is given in Appendix B.5.1. □

This result shows that $G_{01} > G_{02}$ and $G_{11} > G_{12}$, which justifies Remark 3. Therefore, in Problem 3, we can restrict our attention to the maximisation of $G_{01}$ and $G_{11}$. The result also shows that when $E_{11}$ and $E_{01}$ are both defined, then we have $G_{11} > G_{01}$. Table A6 shows that both $E_{11}$ and $E_{01}$ exist simultaneously only in regions $\mathcal{I}_6$, $\mathcal{I}_7$, and $\mathcal{I}_8$, and that in this case $E_{11}$ is stable while $E_{01}$ is unstable. However, it is possible for one to be defined without the other being defined, as shown in Table A6. Indeed, in the regions $\mathcal{I}_1$ and $\mathcal{I}_2$, $E_{01}$ exists and is stable, while $E_{11}$ does not exist and in the regions $\mathcal{I}_4$ and $\mathcal{I}_5$, $E_{11}$ exists and is stable, while $E_{01}$ does not exist. Therefore, the maximum of $G_{11}$ and $G_{01}$ can be obtained for different values of the dilution rate $D$, and the last part of Problem 3 is to fix $S_1^{in}$ and $S_2^{in}$ and compare

$$\max_D G_{01}(D, S_2^{in}) \quad \text{and} \quad \max_D G_{11}(D, S_1^{in}, S_2^{in}).$$

3.2.2. Best Operating Conditions for $G_{01}$ and $G_{11}$

Let us fix the operating parameters $S_1^{in}$ and $S_2^{in}$. We restrict our attention to the case $a_1 = a_2 = 0$ and $\alpha_1 = \alpha_2 = \alpha$, which was considered in [11]. The general case can be considered without added difficulty. Our aim is to compute the values of $D$ for which the functions

$$D \mapsto G_{01}(D, S_2^{in}) \quad \text{and} \quad D \mapsto G_{11}(D, S_1^{in}, S_2^{in})$$

reach their maxima. These functions are proportional to the functions

$$G_0(D) = D\left(S_2^{in} - \lambda_2(\alpha D)\right) \tag{34}$$

$$G_1(D) = D\left(S_2^{in} + \frac{k_2}{k_1}S_1^{in} - \lambda_2(\alpha D) - \frac{k_2}{k_1}\lambda_1(\alpha D)\right) \tag{35}$$

respectively, where $\lambda_1$ and $\lambda_2$ are defined in Table 1. Therefore, $G_{01}$ has an absolute maximum if $G_0$ has one, and this maximum is reached at the same point where $G_0$ reaches its maximum. Similarly, $G_{11}$ has an absolute maximum if $G_1$ has one, and this maximum is reached at the same point where $G_1$ reaches its maximum. To obtain the maximum of $G_0$, we differentiate $G_0$ with respect to $D$. The derivative is given by

$$G_0'(D) = S_2^{in} - \gamma_2(\alpha D)$$

where $\gamma_2$ is defined by

$$\gamma_2(D) = \lambda_2(D) + D\lambda_2'(D). \tag{36}$$

Similarly, the derivative of $G_1$ is given by

$$G_1'(D) = S_2^{in} - \gamma_2(\alpha D) + \frac{k_2}{k_1}\left(S_1^{in} - \gamma_1(\alpha D)\right)$$

where $\gamma_1$ is defined by

$$\gamma_1(D) = \lambda_1(D) + D\lambda_1'(D). \tag{37}$$

**Remark 10.** *Using $\lambda_1'(D) = 1/\mu_1'(\lambda_1(D))$ and $\lambda_2'(D) = 1/\mu_2'(\lambda_2(D))$, the functions $\gamma_2$ and $\gamma_1$ can be written*

$$\gamma_2(D) = \lambda_2(D) + \frac{D}{\mu_2'(\lambda_2(D))}, \quad \gamma_1(D) = \lambda_1(D) + \frac{D}{\mu_1'(\lambda_1(D))}.$$

We make the following assumptions:

**Hypothesis 6.** *The function $\gamma_2 : I_2 \to (0 + \infty)$, defined on $I_2 = (0, \mu_2(S_2^m))$ by (36), is $\mathcal{C}^1$, and for all $D \in I_2$ we have $\gamma_2'(D) > 0$.*

**Hypothesis 7.** *The function $\mu_1 : I_1 \to (0 + \infty)$, defined on $I_1 = (0, m_1)$ by (37), is $\mathcal{C}^1$ and for all $D \in I_1$ we have $\gamma_1'(D) > 0$.*

If Hypothesis 6 is satisfied, then the function $\gamma_2$ is invertible, and for each $S_2^{in}$, the equation

$$S_2^{in} = \gamma_2(\alpha D) \tag{38}$$

has a unique solution, denoted

$$D_0^*(S_2^{in}) = \frac{1}{\alpha}\gamma_2^{-1}(S_2^{in}), \tag{39}$$

where $\gamma_2^{-1}$ is the inverse function of $\gamma_2$. On the other hand, if Hypotheses 6 and 7 are satisfied, the function $\gamma_2 + \frac{k_2}{k_1}\gamma_1$ is $\mathcal{C}^1$ and increasing, since it is the sum of two increasing functions. Therefore, for each $S_1^{in}$ and $S_2^{in}$, the equation

$$S_2^{in} + \frac{k_2}{k_1}S_1^{in} = \gamma_2(\alpha D) + \frac{k_2}{k_1}\gamma_1(\alpha D) \tag{40}$$

has a unique solution, denoted

$$D_1^*(S_1^{in}, S_2^{in}) = \frac{1}{\alpha}\gamma^{-1}\left(S_2^{in} + \frac{k_2}{k_1}S_1^{in}\right), \tag{41}$$

where $\gamma^{-1}$ is the inverse function of $\gamma := \gamma_2 + \frac{k_2}{k_1}\gamma_1$.

The following result gives the answer to the first part of Problem 3.

**Proposition 6.** *Assume that Hypotheses 2, 3, 6, and 7 are satisfied. Then $G_{01}(D, S_2^{in})$ reaches its maximum at $D_0^*(S_2^{in})$, defined by (39) and $G_{11}(D, S_2^{in}, S_2^{in})$ reaches its maximum at the right-hand end of its defining interval, or at $D_1^*(S_1^{in}, S_2^{in})$, defined by (41).*

**Proof.** The proof is given in Appendix B.5.2.  □

The set of best operating conditions for biogas production at $E_{01}$ is the surface $\Gamma_0$ of SOP, defined by:

$$\Gamma_0 = \left\{(D, S_1^{in}, S_2^{in}) : S_2^{in} = \gamma_2(\alpha D)\right\} = \left\{(D, S_1^{in}, S_2^{in}) : D = D_0^*(S_2^{in})\right\} \tag{42}$$

It is the set of operating conditions that produce the maximum of $G_{01}$. The set of best operating conditions for biogas production at $E_{11}$ is the surface $\Gamma_1$ of SOP, defined by:

$$\Gamma_1 = \left\{(D, S_1^{in}, S_2^{in}) : S_2^{in} + \frac{k_2}{k_1}S_1^{in} = \gamma(\alpha D)\right\} = \left\{(D, S_1^{in}, S_2^{in}) : D = D_1^*(S_1^{in}, S_2^{in})\right\} \tag{43}$$

This is the set of operating conditions which produce the maximum of $G_{11}$.

We plot the sets $\Gamma_0$ and $\Gamma_1$ in the 2-dimensional ODs in the $(D, S_1^{in})$-plane shown in Figure 5. Since $S_2^{in}$ is fixed, the set $\Gamma_0$, in blue in the figures, is the vertical line $D = D_0^*(S_2^{in})$,

while $\Gamma_1$, in red in the figures, is the curve of equation $S_1^{in} = \frac{k_1}{k_2}\left(\gamma(\alpha D) - S_2^{in}\right)$. Let $S_1^{in}$ and $S_2^{in}$ be fixed. Consider the OD for which $S_2^{in}$ is equal to the fixed value considered and look for the intersections of $\Gamma_0$ and $\Gamma_1$ with the horizontal line where $S_1^{in}$ is kept constant at the fixed value considered. The abscissas of these intersections are the optimal dilution rates $D_0^*\left(S_2^{in}\right)$ and $D_1^*\left(S_2^{in}\right)$ defined by (39) and (41), respectively.

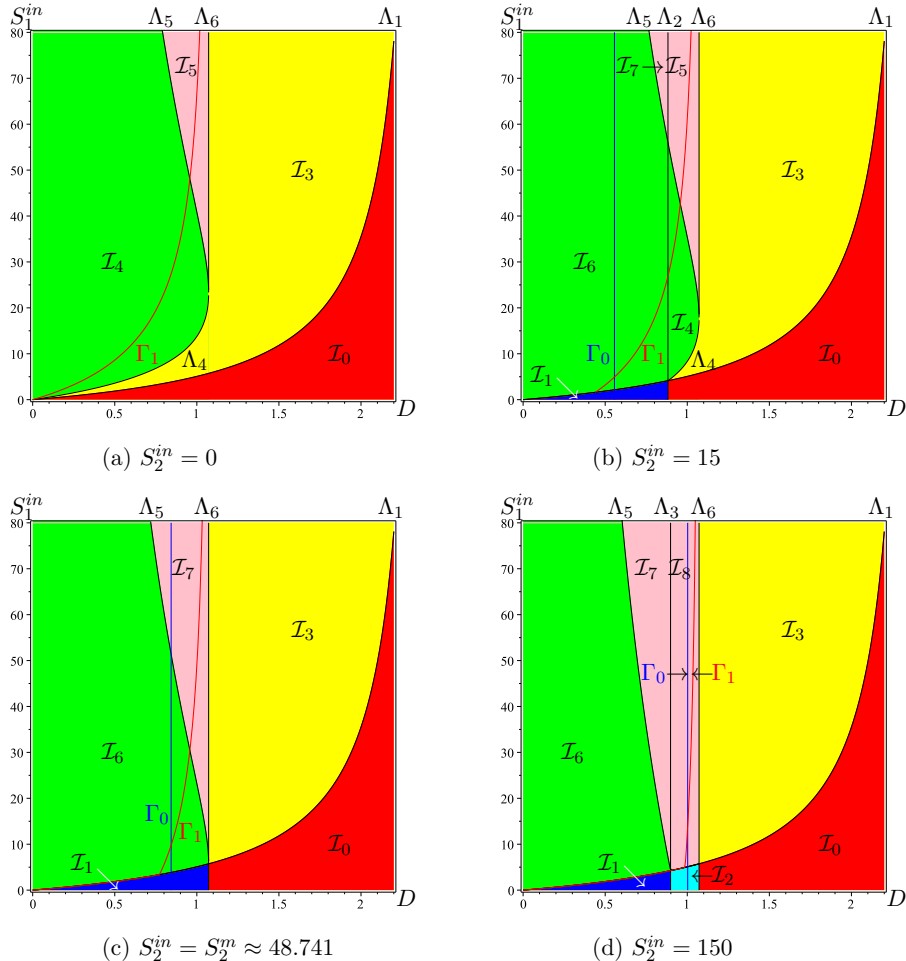

**Figure 5.** The 2-dimensional OD in $\left(D, S_1^{in}\right)$, obtained by cuts at $S_2^{in}$ constant of the 3-dimensional OD shown in Figure A6. The curve $\Gamma_1$, in red, is the set of maximisation of $G_{11}$. The vertical line $\Gamma_0$, in blue, is the set of maximisation of $G_{01}$.

**Remark 11.** *As for the one-step model with a Haldane type growth function, shown in Figure 1b, there exists a threshold value $S_1^c$ corresponding to the intersection point $(D^c, S_1^c)$ of curves $\Gamma_1$ and $\Lambda_5$, such that, if $S_1^{in} > S_1^c$, then the best operating point lies in the bistability pink region; see Figure 6a. The value $D = D^c$ is the solution of equation*

$$\bar{\lambda}_2(\alpha D) = \lambda_2(\alpha D) + \alpha D \lambda_2'(\alpha D) + \frac{k_2}{k_1}\alpha D \lambda_1'(\alpha D), \tag{44}$$

*which gives the abscissa of the point of intersection of $\Gamma_1$ and $\Lambda_5$, and $S_1^c$ is given by*

$$S_1^c = \lambda_1(\alpha D^c) + \frac{k_1}{k_2}\left(\bar{\lambda}_2(\alpha D^c) - S_2^{in}\right).$$

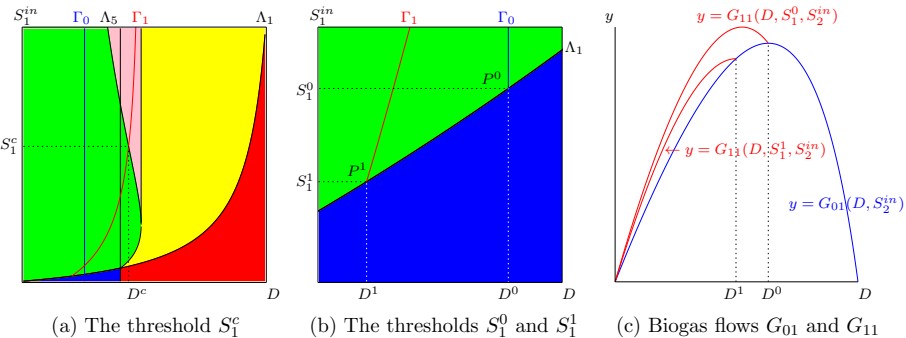

(a) The threshold $S_1^c$     (b) The thresholds $S_1^0$ and $S_1^1$     (c) Biogas flows $G_{01}$ and $G_{11}$

**Figure 6.** (**a**): The point $(D^c, S_1^c) = \Gamma_1 \cap \Lambda_5$. (**b**): A zoom showing the points $P^0 = \Gamma_0 \cap \Lambda_1$ and $P^1 = \Gamma_1 \cap \Lambda_1$. (**c**): The function $D \mapsto G_{01}(D, S_2^{in})$ in blue and the functions $D \mapsto G_{11}(D, S_1^{in}, S_2^{in})$, in red, for $S_1^{in} = S_1^0$ and $S_1^{in} = S_1^1$.

### 3.2.3. The Maximum of $G_{01}$ Can Be Larger than the Maximum of $G_{11}$

In addition to the threshold $S_1^c$, Figures 5 and 6 show two other thresholds obtained by considering the intersection of the $\Gamma_0$ and $\Gamma_1$ curves with the $\Lambda_1$ curve. We depict in Figure 6 a typical situation and show in a zoom the points of intersection. Let $P^0 = (D^0, S_1^0)$ be the point of intersection of $\Gamma_0$ with $\Lambda_1$; see Figure 6b. If $S_1^{in} = S_1^0$, then the productivity $G_{11}$ is defined for $0 \le D \le D^0$ and reaches its maximum for some $D_1^*(S_1^0, S_2^{in}) < D^0$. Moreover, we have

$$\max_D G_{01}(D, S_2^{in}) = G_{01}(D^0, S_2^{in}) = G_{11}(D^0, S_1^0, S_2^{in}).$$

Therefore, see Figure 6c, we have

$$\max_D G_{11}(D, S_1^0, S_2^{in}) > \max_D G_{01}(D, S_2^{in}).$$

Since the function $S_1^{in} \mapsto G_{11}(D, S_1^{in}, S_2^{in})$ is increasing, the same result is true for any $S_1^{in} > S_1^0$. Note that $S_1^0$ depends on $S_2^{in}$ and is a solution of the set of equations

$$S_1^{in} = \lambda_1(\alpha D), \qquad S_2^{in} = \gamma_2(\alpha D)$$

which give the point of intersection of $\Lambda_1$ and $\Gamma_0$. Therefore, $(S_1^0, S_2^{in})$ belongs to the curve

$$\Sigma_0 = \left\{ (S_1^{in}, S_2^{in}) : S_2^{in} = \gamma_2\big(\mu_1(S_1^{in})\big) \right\}. \tag{45}$$

Similarly, let $P^1 = (D^1, S_1^1)$ be the point of intersection of the curves $\Gamma_1$ and $\Lambda_1$; see Figure 6b. If $S_1^{in} = S_1^1$ then the productivity $G_{11}$ is defined for $0 \le D \le D^1$ and reaches its maximum for $D = D^1$. Since $D^1 < D^0$, we have (see Figure 6c),

$$G_{11}(D^1, S_1^1, S_2^{in}) = G_{01}(D^1, S_2^{in}) < G_{01}(D^0, S_2^{in}).$$

Therefore,

$$\max_D G_{11}(D, S_1^1, S_2^{in}) < \max_D G_{01}(D, S_2^{in}).$$

The same result is true for any $S_1^{in} < S_1^1$, because the function $S_1^{in} \mapsto G_{11}(D, S_1^{in}, S_2^{in})$ is increasing. Note that $S_1^1$ depends on $S_2^{in}$ and is a solution to the set of equations

$$S_1^{in} = \lambda_1(\alpha D), \qquad S_2^{in} + \frac{k_2}{k_1} S_1^{in} = \gamma_2(\alpha D) + \frac{k_2}{k_1} \gamma_1(\alpha D),$$

which give the point of intersection of $\Lambda_1$ and $\Gamma_1$. Therefore $(S_1^1, S_2^{in})$ belongs to the curve

$$\Sigma_1 = \left\{ (S_1^{in}, S_2^{in}) : S_2^{in} = \sigma_1(S_1^{in}) \right\}, \quad \text{where} \quad \sigma_1(S_1^{in}) = \gamma_2\big(\mu_1(S_1^{in})\big) + \frac{k_2}{k_1} \frac{\mu_1(S_1^{in})}{\mu_1'(S_1^{in})}. \tag{46}$$

The curves $\Sigma_0$ and $\Sigma_1$ are illustrated in Figure 7b. We have the following result.

**Proposition 7.** *Let $\Sigma_0$ and $\Sigma_1$ be the curves of the $(S_1^{in}, S_2^{in})$ plane defined by (45) and (46), respectively. If $(S_1^{in}, S_2^{in})$ is at the right of $\Sigma_0$, then we have*

$$\max_D G_{11}(D, S_1^1, S_2^{in}) > \max_D G_{01}(D, S_2^{in}).$$

*If the function $\mu_1 / \mu_1'$ is increasing and $(S_1^{in}, S_2^{in})$ is at the left of $\Sigma_1$, then we have*

$$\max_D G_{11}(D, S_1^1, S_2^{in}) < \max_D G_{01}(D, S_2^{in}).$$

**Proof.** The proof is given is Appendix B.5.3.  □

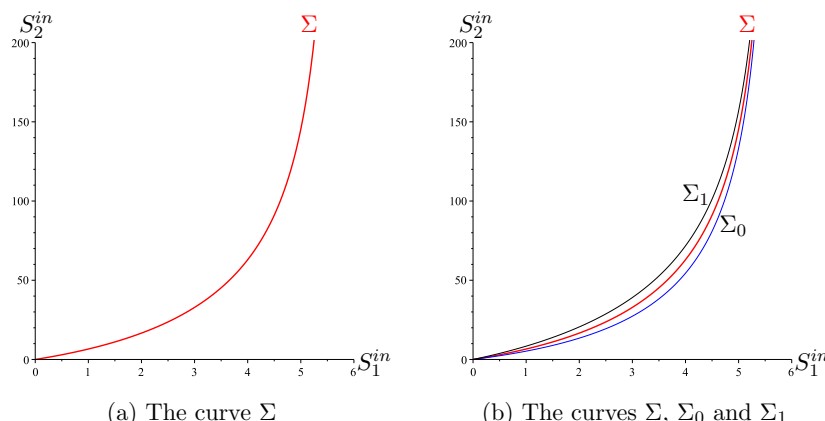

(a) The curve $\Sigma$          (b) The curves $\Sigma$, $\Sigma_0$ and $\Sigma_1$

**Figure 7.** To the left of the curve $\Sigma$ we have $\max_D G_{01} > \max_D G_{11}$ and to its right we have $\max_D G_{01} < \max_D G_{11}$.

Now, we give the curve $\Sigma$ lying between the $\Sigma_0$ and $\Sigma_1$ curves, such that the maximum of biogas flow rate is obtained for $E_{01}$ at the left of $\Sigma$ and for $E_{11}$ at the right of $\Sigma$; see Figure 7a. We need the following hypothesis.

**Hypothesis 8.** *We assume that the function $\phi$ defined by $\phi(D) = D^2 \lambda_2'(D)$ is increasing.*

Therefore, $\phi$ has an inverse function $\phi^{-1}$ defined by $D = \phi^{-1}(B)$ if and only if $D$ is the solution to equation $\phi(D) = B$. Consider the curve $\Sigma$ defined by the parametric equations

$$S_2^{in} = \gamma_2(\Delta(D)), \quad S_1^{in} = \gamma_1(\alpha D) + \tfrac{k_1}{k_2}(\gamma_2(\alpha D) - \gamma_2(\Delta(D))) \tag{47}$$

where $\Delta(D)$ is defined by

$$\Delta(D) := \phi^{-1}\left(\alpha^2 D^2 \left(\lambda_2'(\alpha D) + \tfrac{k_2}{k_1}\lambda_1'(\alpha D)\right)\right). \tag{48}$$

The following result gives the answer to the second part of Problem 3.

**Proposition 8.** *Assume that Hypothesis 8 is satisfied and, in addition, the curve $\mathcal{C}$ defined by the parametric Equation (47) is the graph of an increasing function $S_2^{in} \mapsto S_1^{in}$. Then it is the subset of the $(S_1^{in}, S_2^{in})$ plane, where*

$$\max_D G_{01}(D, S_2^{in}) = \max_D G_{11}(D, S_1^{in}, S_2^{in}). \tag{49}$$

*To the left of $\mathcal{C}$, we have $\max_D G_{01} > \max_D G_{11}$ and to its right, we have $\max_D G_{01} < \max_D G_{11}$.*

**Proof.** The proof is given in Appendix B.5.4. □

**Remark 12.** *By combining the result of Remark 11 with that of Proposition 8, we deduce that the curve $\Sigma$ and the straight line $\mathcal{C}$ defined by*

$$\mathcal{C} := \left\{ (S_1^{in}, S_2^{in}) : S_2^{in} + \tfrac{k_2}{k_1} S_1^{in} < \gamma_2(\alpha D^c) + \tfrac{k_2}{k_1}\gamma_1(\alpha D^c) \right\},$$

*where $D = D^c$ is the solution of Equation (44), divide the plane $(S_1^{in}, S_2^{in})$ into three regions:*

$$\begin{aligned}
\mathcal{R}_0 &:= \left\{ (S_1^{in}, S_2^{in}) \text{ lies to the left of } \Sigma \right\} \\
\mathcal{R}_1 &:= \left\{ (S_1^{in}, S_2^{in}) \text{ lies to the right of } \Sigma \text{ and to the left of } \mathcal{C} \right\} \\
\mathcal{R}_2 &:= \left\{ (S_1^{in}, S_2^{in}) \text{ lies to the right of } \Sigma \text{ and } \mathcal{C} \right\}.
\end{aligned}$$

*In the region $\mathcal{R}_0$, we have $\max_D G_{10} > \max_D G_{11}$. In the region $\mathcal{R}_1$, we have $\max_D G_{10} < \max_D G_{11}$, and the optimal dilution rate corresponds to the global asymptotic stability of $E_{11}$. In the region $\mathcal{R}_2$, we also have $\max_D G_{10} < \max_D G_{11}$, but the optimal dilution rate corresponds to the bistability of $E_{11}$ and $E_{10}$.*

Since the steady state $E_{10}$ does not produce biogas, if the bioreactor is operated in the $\mathcal{R}_2$ region, care should be taken to initialise it in the basin of attraction of $E_{11}$ and not in the basin of $E_{10}$. The regions are illustrated in Figure 8a, obtained with the parameter values given in Table A8. Let us illustrate the behaviour of $G_{01}(D, S_2^{in})$ and $G_{11}(D, S_1^{in}, S_2^{in})$, as functions of $D$, for the operating points $o_k \in \mathcal{R}_k$, $k = 0, 1, 2$, shown in Figure 8a. Figure 8b shows the OD in the $(D, S_1^{in})$ plane and $S_2^{in} = 15$. The horizontal lines $S_1^{in} = 1.5, 10$, and 50, corresponding to the points $o_0 = (1.5, 15)$, $o_1 = (10, 15)$, and $o_2 = (50, 15)$, respectively, give the optimal dilution rates. For $o_0$, the maximum of the biogas flow is obtained for $E_{01}$; see Figure 8c. For $o_1$, the maximum of the biogas flow is obtained for $E_{11}$, and $E_{11}$ is GAS; see Figure 8d. For $o_2$, the maximum of the biogas flow is obtained for $E_{11}$, but $E_{11}$ is only LAS; see Figure 8e.

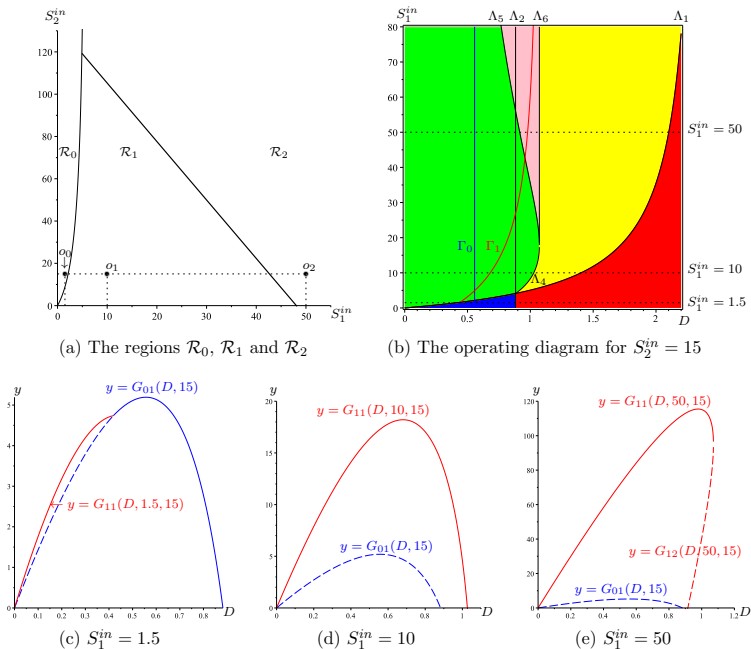

(a) The regions $\mathcal{R}_0$, $\mathcal{R}_1$ and $\mathcal{R}_2$

(b) The operating diagram for $S_2^{in} = 15$

(c) $S_1^{in} = 1.5$

(d) $S_1^{in} = 10$

(e) $S_1^{in} = 50$

**Figure 8.** The biogas flow rates $D \mapsto G_{01}(D, S_2^{in})$ in blue, $D \mapsto G_{11}(D, S_1^{in}, S_2^{in})$ in red, and $D \mapsto G_{12}(D, S_1^{in}, S_2^{in})$ in dashed red, corresponding to the operating points (**c**) $o_0 = (1.5, 15)$, (**d**) $o_1 = (10, 15)$ and (**e**) $o_2 = (50, 15)$. The flow rate biogas of a stable steady state is drawn in bold, while it is drawn in dashed line when the steady state is unstable.

### 3.2.4. Applications to the Classical AM2 Model

The dynamical equations of the model are

$$
\begin{aligned}
\dot{S}_1 &= D\big(S_1^{in} - S_1\big) - k_1\mu_1(S_1)X_1, \\
\dot{X}_1 &= (\mu_1(S_1) - \alpha D)X_1, \\
\dot{S}_2 &= D\big(S_2^{in} - S_2\big) + k_2\mu_1(S_1)X_1 - k_3\mu_2(S_2)X_2, \\
\dot{X}_2 &= (\mu_2(S_2) - \alpha D)X_2,
\end{aligned}
\tag{50}
$$

where the kinetics $\mu_1$ and $\mu_2$ are given by

$$
\mu_1(S_1) = \frac{m_1 S_1}{K_1 + S_1}, \qquad \mu_2(S_2) = \frac{m_2 S_2}{K_2 + S_2 + S_2^2/K_i},
\tag{51}
$$

For the Monod and Haldane functions, Hypotheses 2 and 3 are satisfied and the break-even concentrations can be calculated explicitly. For the convenience of the reader we summarize in Table 4 the expressions of the break even concentrations and the auxiliary functions that are needed in the description of the results. The OD in the three dimensional SOP, corresponding to the biological value parameters given in Table A8 is shown in Figure A6 of the Appendix. The two-dimensional diagrams in the $(D, S_1^{in})$ plane, where $S_2^{in}$ is kept constant, are depicted in Figure A7. The two-dimensional diagrams in the $(S_1^{in}, S_2^{in})$ plane, where $D$ is kept constant, are depicted in Figure A8.

**Table 4.** Auxiliary function in the classical AM2 model.

| $\mu_1(S_1) = \frac{m_1 S_1}{K_1 + S_1}$ | |
|---|---|
| $\lambda_1(D) = \frac{DK_1}{m_1 - D}$, | Defined for $0 \le D < m_1 = \mu_1(+\infty)$ |
| $\mu_2(S_2) = \frac{m_2 S_2}{K_2 + S_2 + S_2^2/K_i}$, | $S_2^m = \sqrt{K_2 K_i}, \mu_2(S_2^m) = \frac{m_2}{1 + 2\sqrt{K_2/K_i}}$ |
| $\lambda_2(D) = \frac{(m_2 - D) - \sqrt{(m_2 - D) - 4D^2 K_2/K_i}}{2D} K_i$, | Defined for $0 < D \le \mu_2(S_2^m)$ |
| $\bar{\lambda}_2(D) = \frac{(m_2 - D) + \sqrt{(m_2 - D)^2 - 4D^2 K_2/K_i}}{2D} K_i$ | Defined for $0 < D \le \mu_2(S_2^m)$ |
| $\gamma_1(D) = \lambda_1(D) + D\lambda_1'(D)$, | Defined for $D < m_1$ |
| $\gamma_2(D) = \lambda_2(D) + D\lambda_2'(D)$, | Defined for $D < \mu_2(S_2^m)$ |
| $\gamma(D) = \gamma_2(D) + \frac{k_2}{k_1}\gamma_1(D)$ | Defined for $D < \min(m_1, \mu_2(S_2^m))$ |

Since $\mu_1'' < 0$ and $\mu_2'' < 0$ on $(0, S_2^m)$, from Lemma 3 we deduce that $\gamma_1' > 0$ and $\gamma_2' > 0$. Therefore Hypotheses 6 and 7 are satisfied. From Proposition 6, we deduce that the curves $\Gamma_0$ and $\Gamma_1$, defined by (42) and (43) are the sets of best operating conditions for $G_{01}$ and $G_{11}$, respectively. These sets are shown in Figure 5, for some of the ODs depicted in Figure A7.

On the other hand, since $\lambda_2'' > 0$, we deduce that $\phi' > 0$, where $\phi(D) = D^2\lambda_2'(D)$. Hence Hypothesis 8 is satisfied. The inverse function of $\phi$ can be computed explicitly. We have

$$
\phi^{-1}(B) = m_2 \frac{(m_2 K_i + 2B)\sqrt{BK_2 K_i(m_2 K_i + B)} - (m_2 K_i + B)K_i B}{K_2 m_2^2 K_i^2 + (4K_2 - K_i)(m_2 K_i + B)B}
$$

Note that the function $\mu_1/\mu_1'$ is increasing. Therefore, the result of Proposition 7 is true. Straightforward computation shows that the curve $\Sigma$ is increasing. Hence, the result of Proposition 8 is true. The curve $\Sigma$ of the $(S_1^{in}, S_2^{in})$-plane,

$$
\max_D G_{01}(D, S_2^{in}) = \max_D G_{11}(D, S_1^{in}, S_2^{in}),
$$

and the curves $\Sigma_0$ and $\Sigma_1$ are shown in Figure 7. Finally the regions $\mathcal{R}_0$, $\mathcal{R}_1$, and $\mathcal{R}_2$ and the behaviour of the biogas flow rates $D \mapsto G_{01}(D, S_2^{in})$ and $D \mapsto G_{11}(D, S_1^{in}, S_2^{in})$ are depicted in Figure 8 for three operating points $o_j \in \mathcal{R}_j$, $j = 0, 1, 2$.

### 3.3. Relationship with Previous Results

The OD of the one-step model is well known in the existing literature [22,36]. In these references, the dilution rate is shown on the vertical axis, and the input substrate concentration is shown on the horizontal axis. In this paper, we have reversed the axes, because, as we then consider the biogas flow rate, or productivity, as a function of the dilution rate, it is interesting to have the dilution rate on the horizontal axis in all graphs.

In practical applications, when maximising biogas or biomass production, the substrate concentration $S^{in}$ is given and the optimal dilution rate $D^*(S^{in})$, depending on $S^{in}$, that maximises biogas or biomass production must be determined. For the Monod function, the formula giving the optimal dilution appears in several reference books; see for example Formula (13.70) in [12] or Formula (6.83) in [19]. For the Monod and Haldane functions, it appears in [20,21] and were used for the optimization of bioreactors by extremum seeking. The approach used here is to try to directly exploit the equation of which the optimal $D$ is a solution and to represent its solutions in the OD. To the best of our knowledge, the set of best operating conditions for biogas or bimass production have only recently been drawn in the OD [51–53]. In these papers the main problem is to consider the optimisation of biogas flow rate or biomass productivity in the serial chemostat and to compare the performances of the serial chemostat with a single chemostat of the same total volume.

In the case without biomass mortality, the mathematical analysis of the two-step model was given in [15], in the case $\alpha = 1$, and in [26] in the case $\alpha \leq 1$. The OD was given in [42]. Here we have extended these results to the case including mortality. The maximization of biogas flow for this model has been well studied in [11]. For example, the curves $\Sigma_0$ and $\Sigma_1$ were described( see Figure 4 in [11]), where the curves are called $C_2$ and $C_3$, respectively. The existence of the curve $\Sigma$ was predicted; see Remark 7 in [11]. However, neither its analytical equation nor its numerical representation was given in [11]. Note that the curves $\Sigma_0$ and $\Sigma_1$ have vertical asymptotes; see Figure 7. We deduce that the curve $\Sigma$ also has one. Therefore, the region $\mathcal{R}_0$ is not wide. That is to say, for an $S_2^{in}$ as large as one wants, it is enough that $S_1^{in}$ exceeds a certain threshold, corresponding to the vertical asymptote, for the system to be in the $\mathcal{R}_1$ or $\mathcal{R}_2$ region.

The representation of the set of optimal operating conditions in the OD, as well as its use to deduce the various properties of biogas production, is not found in the existing literature. In particular, the identification of the threshold at which the system will operate in a bistability regime is new and answers practical questions of great interest for bioreactors and their management. These questions are related to the so-called stability criteria named "overloading tolerance" or "destabilization risk index" [26,56]. This index alerts the experimenter as soon as the system approaches a regime of bistability. Bistability in the model occurs when the unstable steady states $E_{02}$ or $E_{12}$ exist. For example, although the steady state $E_{12}$ is unstable, if it exists, its existence completely changes the functioning of the system. Indeed, in this case, the steady state $E_{10}$, of washout of the methanogenic bacteria (without biogas production), becomes stable, and the positive steady state $E_{11}$ loses its global stability. This important issue is not addressed in [11], where the authors do not consider the steady states $E_{02}$ and $E_{12}$. They justify their disregard by the fact that these steady states are unstable, that their biogas flow rate is lower than the biogas flow rate of the associated steady states $E_{01}$ and $E_{11}$, and also because according to them their conditions of existence are the same as those of the steady states $E_{01}$ and $E_{11}$; see Section 3 in [11]. The first two reasons are of course correct, but the third is not. Indeed, $E_{11}$ can exist without $E_{12}$ existing. On the other hand, when $E_{12}$ exists, $E_{11}$ must also exist, and we have the phenomenon of bistability of $E_{10}$ and $E_{11}$. In this paper, we considered all steady states, which allowed us to highlight the important region of bistability (coloured in pink in the figures) and thus to provide a valuable tool for the experimenter to avoid monitoring the system in this region, or at least to be very careful if he should do so.

## 4. Discussion

In this paper, we have determined the set of operating parameters that optimise the biogas flow in simple AD models. We have represented these sets in the OD of the model. This representation allowed us to obtain a simple graphic visualisation of the optimal operating conditions. It also allows direct discovery of the properties of these optimal conditions.

To illustrate the simplicity with which the properties appear in the OD, let us consider the case with inhibition by the substrate when its concentration is high (Haldane function). It is well known that when the inflowing substrate concentration of the bioreactor is high, the system presents bistability, with a risk of convergence towards the washout steady state. It is natural then to ask whether operating conditions that maximise the biogas flow can lead to this bistability situation. This phenomenon was already observed, using the OD, in a more complex system [35]. The main result of this paper is to address this problem and to give a complete answer in one-step and two-steps models. Although we have an explicit formula for the optimal dilution rate as a function of the substrate input concentration, this formula does not allow us to easily determine whether or not the system is in the instability zone. On the other hand, drawing the set of optimal conditions in the OD immediately shows that this set enters the bistability zone and allows to find the critical threshold of the substrate input concentration at which the system will operate in the instability zone; see the threshold $S^c$ in Figure 1b. This shows the value of the OD in understanding the model.

The contribution of the OD to the understanding of the system's behaviour is even more spectacular in the case of the AM2 model. In this case, there are three operating parameters, and the OD must be represented in the plane formed by two of them by fixing the third. The role of this third parameter is described by a series of diagrams. The sets of optimal operating conditions are surfaces in the space of the three operating parameters, whose traces in the two dimensional ODs are curves. It is immediately apparent whether these curves fall within the areas where the system behaviour may be at risk and the thresholds can be easily found. Three regions can then be determined in the plane of the concentrations of the two input substrates. In one of the regions, the maximum biogas flow rate of the steady state where both acidogenic and methanogenic bacteria are present is reached for a value of the dilution ratio for which the acidogenic bacteria are washed out. In a second region, the maximum is reached for a value of the dilution rate for which the positive steady state is GAS. In a third region, the maximum is reached for a value of the dilution rate for which the system presents à bistability behaviour; see Figure 8. These regions have not been identified in the existing literature.

Some figures in this paper (see Figures 1–4, 6, A4, and A5) are made without graduations on the axes because they represent generic situations where the growth functions verify our general hypothesis and the biological parameters are not specified. However, in practice, to construct an OD, one fixes the growth functions and biological parameters and then draws the curves separating the regions of the OD. Indeed, the OD is a tool for the experimenter who knows the biological parameter values of the model he is considering, and then plots its OD. We do that in Appendix A.5 for some classical growth functions; see Figures A1–A3. See also Figures 5, 7 and 8 in Section 3.2, for the AM2 model, whose biological parameters are given in Table A8. See also Figures A6–A8 in the Appendix B.

Another result obtained with the help of the OD of a two-step model is worth mentioning here. It was shown in [42,57] that under certain circumstances, increasing the dilution rate can globally stabilize two-step biological systems. This kind of surprising and unexpected result was obtained also for a two-step model where the first reaction has a Contois kinetics instead of a Monod one [58]. These studies have shown how unexpected properties can be discovered and studied by analysing the OD of the model. Our findings in this paper are a further illustration of the relevance of the OD in the study of one-step and two-step models.

The two-step models of the form (12) present a commensalistic relationship between microorganisms. For definitions and complementary information on commensalism,

the reader may consult [59]. Methanogenic bacteria use for their growth the product of the acidogenic bacteria, but acidogenic bacteria are not affected by the growth of the methanogenic bacteria. More complex models are those studied in [38–40,43], which present a syntrophic relationship between the micro organisms: the first population is affected by the growth of the second population. For more details and information on commensalism and syntrophy, the reader is referred to [40,43,59–64] and the references therein. The ODs of some of these models are well understood; see [38–40,43]. Studying the biogas or biomass production for these more complex and more realistic models of AD is a challenging question. It is the subject for future research directions. The determination of the OD and the optimal productivity of synthetic microbial communities considered in [65] is also an interesting question that deserves further attention.

## 5. Conclusions

In this work, we considered one-step and two-step simple models of AD which are able to adequately capture the main dynamical behaviour of the full ADM1 and have the advantage that a complete analysis for the existence and local stability of their steady states is available. These models have been validated on real data. We considered that the biological parameters of the models have been calibrated on the data. Therefore, the OD of the model can be constructed, and the results can be illustrated in the OD. The best operating conditions for biogas production or biomass production are obtained as subsets of the OD.

For a one-step model, the set $\Gamma$ of best operating conditions for biogas production is described as a curve of equation $S^{in} = \gamma(D)$; see Figure 2 for the Monod case and Figure 3 for the Haldane case. These curves permit the optimal dilution $D_G^*(S^{in})$ for which the biogas production is maximal to be obtained graphically and easily. The explicit expression for $D_G^*(S^{in})$ is not always available, and even when it is known, see Appendix A.5. On the other hand, the graphical visualisation of $D_G^*(S^{in})$ in the OD allows us to predict the behaviour of the system when it is operated at this optimal dilution rate, as illustrated in Figures 2 and 3 and A1–A3.

When there are no maintenance terms included in the model, it is known that biogas production and biomass production are given by the same expressions. Therefore, the maximum of these quantities is obtained for the same operating conditions. However, when maintenance is included in the model, the subsets of best operating conditions for biogas production and biomass production are not the same; see Figure 4.

For a two-step model, we obtain two subsets, $\Gamma_0$ and $\Gamma_1$, of maximal biogas production, corresponding to the steady states $E_{01}$ and $E_{11}$, respectively; see Figure 5. The steady state $E_{01}$ corresponds to the washout of the first biomass, while $E_{11}$ corresponds to the persistence of the two populations. For certain operating conditions, the biogas production of $E_{01}$ can be higher than that of $E_{11}$. We have determined the set of values for the input substrate concentrations for which this occurs; see Figure 7. We have identified two other subsets of operating conditions in which the system behaves in different ways; see Figure 8. In one set the optimal dilution rate corresponds to an operating regime where the system is functioning at a GAS steady state, while, in the second, there is bistability. It may be in the experimenter's interest to run the system with operating parameters that give rise to bistability, since the biogas flow rate is then greater. However, they must be careful to initialise it in the basin of attraction of the steady state $E_{11}$, because otherwise it may converge towards the steady state $E_{10}$, which does not produce biogas.

Our findings illustrate how the OD is a useful tool for the understanding of the behaviour of one-step and two-step models. The OD can be constructed once the biological parameters of the model are fixed. It can also be constructed qualitatively, without specifying the values of the biological parameters. It is therefore a powerful tool for the mathematical analysis of a model when the growth functions are not specified. It is also a tool that allows us to answer important and natural questions that we might not have asked ourselves without this tool. Therefore, the OD allows new interesting questions to

be asked and answered about the model. When studying any problem concerning the chemostat, it is useful to represent the results obtained in the OD. This gives a very clear overview of the system and its operating modes. In this paper, we have illustrated the effectiveness of this approach in the study of the maximisation of the biogas flow rate and the productivity of the biomass.

**Funding:** This research received no external funding.

**Institutional Review Board Statement:** Not applicable.

**Informed Consent Statement:** Not applicable.

**Data Availability Statement:** Data sharing is not applicable to this article as no datasets were generated or analysed during the current study.

**Acknowledgments:** The author is grateful to Radhouane Fekih-Salem and Jérôme Harmand for insightful comments on this work. The author is also grateful to two anonymous reviewers for comments that greatly improved this manuscript. The author thanks the UNESCO ICIREWARD project ANUMAB and the Euro-Mediterranean research network Treasure (http://www.inrae.fr/treasure) (accessed on 20 January 2022).

**Conflicts of Interest:** The author declares no conflict of interest.

## Abbreviations

The following abbreviations are used in this manuscript:

| | |
|---|---|
| AD | Anaerobic Digestion |
| ADM1 | The IWA Anaerobic Digestion Model No 1, see [2] |
| AM2 | Anaerobic Digestion Model of [25] |
| CSTR | Continuous Stirred Tank Reactor or Bioreactor, or Chemostat |
| GAS | Globally Asymptotically Stable |
| HRT | Hydraulic Retention Time |
| LAS | Locally Asymptotically Stable |
| MBR | Membrane Bioreactor |
| OD | Operating Diagram |
| SOP | Set of Operating Parameters |
| SRT | Solid Retention Time |
| U | Unstable |
| VFA | Volatile Fatty Acids |

## Appendix A. One-Step Model

*Appendix A.1. Model Reduction*

We consider the chemostat model (2). It is usual in mathematical theory [22,24] to make the change of variable $x = kX$, which transforms (2) into

$$
\begin{aligned}
\dot{S} &= D(S^{in} - S) - \mu(S)x \\
\dot{x} &= (\mu(S) - D_1)x
\end{aligned}
$$

Therefore, the stoichiometric coefficient $k$ can be reduced to 1 in (2).

*Appendix A.2. The Operating Diagram of the One-Step Model*

In order to construct the OD of (2), one needs to determine and compute the boundaries of the regions of the diagram, i.e., to compute the parameter values at which a qualitative change in the dynamic behaviour of (2) occurs. For (2), these boundaries are the curves

$$
\begin{aligned}
\Lambda &= \left\{ (D, S^{in}) : S^{in} = \lambda(\alpha D + a) \right\}, \\
\Lambda_2 &= \left\{ (D, S^{in}) : S^{in} = \bar{\lambda}(\alpha D + a) \right\}, \\
\Lambda_1 &= \left\{ (D, S^{in}) : \alpha D + a = \mu(S^m) \text{ and } S^{in} \geq S^m \right\}
\end{aligned}
\tag{A1}
$$

These curves separate the Set of Operating Parameters (SOP)

$$\text{SOP} = \left\{ (D, S^{in}) : D \geq 0 \text{ and } S^{in} \geq 0 \right\},$$

in three regions, denoted $\mathcal{J}_0$, $\mathcal{J}_1$, and $\mathcal{J}_2$, corresponding to different behaviours of (2), as depicted in Table A1.

**Table A1.** Existence and stability of steady states of (2) in the three regions of the operating space. The last column shows the color in which the region is depicted in the OD shown in Figures 1–4 and A1–A3.

| Region | $F_0$ | $F_1$ | $F_2$ | Color |
|---|---|---|---|---|
| $\mathcal{J}_0 = \left\{ (D, S^{in}) : S^{in} \leq \lambda(\alpha D + a) \right\}$ | GAS | | | Yellow |
| $\mathcal{J}_1 = \left\{ (D, S^{in}) : \lambda(\alpha D + a) < S^{in} \leq \bar{\lambda}(\alpha D + a) \right\}$ | U | GAS | | Green |
| $\mathcal{J}_2 = \left\{ (D, S^{in}) : S^{in} > \bar{\lambda}(\alpha D + a) \right\}$ | LAS | LAS | U | Pink |

GAS, LAS, and U mean that the steady state is Globally Asymptotically Stable, Locally Asymptotically Stable, or Unstable, respectively. No letter means that the steady state does not exist in the region. Note that

$$\Lambda \cup \Lambda_2 = \left\{ (D, S^{in}) : D = \frac{\mu(S^{in}) - a}{\alpha} \right\}.$$

We plot in Figure 1 the curves $\Lambda$, $\Lambda_1$, and $\Lambda_2$ in SOP and the regions delimited by these curves. This figure, together with Table A1, is the OD of (2). This diagram is well known in the literature [22,36]. When $S^m = +\infty$, then only $\Lambda$ exists ($\Lambda_1 = \Lambda_2 = \emptyset$). In this case, the OD contains only the regions $\mathcal{J}_0$ and $\mathcal{J}_1$. The main difference between Figure 1a, obtained for the Monod case ($S^m = +\infty$), and Figure 1b, obtained for the Haldane case ($S^m < +\infty$), is the appearance of the region of bistability $\mathcal{J}_2$. In this region, both steady states $F_0$ and $F_1$ are LAS and the asymptotic behaviour of a solution depends on its initial condition. If the initial condition belongs to the basin of attraction of $F_0$, then the species $X$ is washed out from the chemostat. If the initial condition belongs to the basin of attraction of $F_1$, then, when $t \to +\infty$, the concentration $X(t)$ of the species tends to $X^* = \frac{D}{kD_1}(S^{in} - \lambda(D_1))$. The green region $\mathcal{J}_1$ is the "target" operating regions, as it corresponds to the global stability of the steady state, where the species survive. The pink region $\mathcal{J}_2$ corresponds to the bistability of $F_0$ (no biogas production) and $F_1$ (with biogas production). If the chemostat is operated in the region $\mathcal{J}_2$, then, for a good operation of the system, its state at start up should correspond to the convergence toward $F_1$ rather than $F_0$.

*Appendix A.3. Maximization of Biogas Production*

Appendix A.3.1. Proof of Proposition 1

The function $G$ defined by (14) is $\mathcal{C}^1$ on the interior of $I(S^{in})$ and its derivative is given by

$$G'(D) = S^{in} - \gamma(D),$$

where $\gamma$ is defined by (17). Therefore, if $g(S^{in})$ is in the interior of $I(S^{in})$, by Fermat's theorem, any point $D^* \in g(S^{in})$ is a critical point of $G$; i.e., $G'(D^*) = 0$, which is equivalent to $S^{in} = \gamma(D^*)$. The proof of the proposition is complete if we prove that the set $g(S^{in})$ is in the interior of $I(S^{in})$. If $S^{in} < S^m$, then $G$ is defined for $0 \leq D \leq \delta$, where $\delta = \frac{\mu(S^{in}) - a}{\alpha}$, is positive if $0 < D < \delta$ and satisfies $G(0) = 0$ and

$$G(\delta) = \delta(S^{in} - \lambda(\alpha\delta + a)) = \delta(S^{in} - \lambda(\mu(S^{in}))) = \delta(S^{in} - S^{in}) = 0.$$

Therefore, the maximum cannot be attained in 0 or $\delta$. Similarly if $S^m < +\infty$ and $S^{in} \geq S^m$, then $G$ is defined for $0 \leq D \leq \delta$, where $\delta = \frac{\mu(S^m) - a}{\alpha}$, is positive if $0 < D < \delta$ and satisfies $G(0) = 0$ and

$$G(\delta) = \delta(S^{in} - \lambda(\alpha\delta + a)) = \delta(S^{in} - \lambda(\mu(S^m))) = \delta(S^{in} - S^m) \geq 0.$$

Moreover, if $S^{in} > S^m$, we have

$$\lim_{D \to \mu(S^m)} \lambda'(D) = +\infty.$$

Hence,

$$\lim_{D \to \delta} G'(D) = -\infty.$$

Therefore, the maximum cannot be attained in 0 or $\delta$ and $g(S^{in})$ is in the interior of $I(S^{in})$.

### Appendix A.3.2. Proof of Proposition 2

Since $H(\lambda(a)) = H(S^{in}) = 0$ and $H(S) > 0$ for $\lambda(a) < S < S^{in}$, the maximum of $H$ is attained at a point $S^* \in (\lambda(a), S^{in})$. By Fermat's theorem, $S^*$ is a critical point of $H$; i.e., $H'(S^*) = 0$. We have

$$G'(D) = H'(\lambda(\alpha D + a))\lambda'(\alpha D + a).$$

Hence, $H$ has a maximum at $S^*$ if and only if $G$ has a maximum at $D^* = \frac{\mu(S^*) - a}{\alpha}$. The derivative of $H$ is given by

$$H'(S) = \mu'(S)(S^{in} - S) - \mu(S) + a.$$

Hence, $H'(S) = 0$ if and only if $S^{in} = \eta(S)$, where $\eta$ is defined by (21). From $H'(S^*) = 0$, it is deduced that $S^{in} = \eta(S^*)$.

### Appendix A.3.3. Proof of Lemma 1

If $\mu$ is $C^2$, so is $\lambda$ and the derivative of $\gamma$ is given by

$$\gamma'(D) = 2\alpha\lambda'(\alpha D + a) + \alpha^2 D\lambda''(\alpha D + a).$$

Using $\mu(\lambda(D)) = D$, we have

$$\lambda'(D) = \frac{1}{\mu'(\lambda(D))} \quad \text{and} \quad \lambda''(D) = -\frac{\mu''(\lambda(D))\lambda'(D)}{(\mu'(\lambda(D)))^2}. \tag{A2}$$

Hence,

$$\gamma'(D) = \alpha\lambda'(\alpha D + a)\left(2 - \frac{\alpha D\mu''(\lambda(\alpha D + a))}{(\mu'(\lambda(\alpha D + a)))^2}\right).$$

Since $\lambda' > 0$ it is deduced that $\gamma'(D) > 0$ if and only if for all $D \in (0, \delta(S^m))$,

$$\alpha D\mu''(\lambda(\alpha D + a)) < 2(\mu'(\lambda(\alpha D + a)))^2.$$

Using the change of variable $S = \lambda(\alpha D + a)$, this condition is equivalent to: for all $S \in (\lambda(a), S^m)$,

$$(\mu(S) - a)\mu''(S) < 2(\mu'(S))^2.$$

Therefore (1) $\Leftrightarrow$ (2). Moreover, we have

$$\left(\frac{1}{\mu - a}\right)' = -\frac{\mu'}{(\mu - a)^2}, \qquad \left(\frac{1}{\mu - a}\right)'' = \frac{2(\mu')^2 - (\mu - a)\mu''}{(\mu - a)^3}.$$

Hence, $(1/(\mu - a))'' > 0$ if and only if $(\mu - a)\mu'' < 2(\mu')^2$. Therefore (2) $\Leftrightarrow$ (3). The derivative of $\eta$ is given by

$$\eta'(S) = 2 - \frac{(\mu(S)-a)\mu''(S)}{(\mu'(S))^2}.$$

Therefore (2) $\Leftrightarrow$ (4). If $\mu'' < 0$ on $(0, S^m)$, then since $\mu' > 0$ on $(\lambda(a), S^m)$, the condition $(\mu(S) - a)\mu''(S) < (\mu'(S))^2$ is obviously satisfied.

*Appendix A.4. Maximization of Biomass Production*

Appendix A.4.1. Proof of Proposition 3

The function $P$ defined by (23) is $\mathcal{C}^1$ on the interior of $I(S^{in})$, and its derivative is given by

$$P'(D) = \frac{D(\alpha D+2a)}{(\alpha D+a)^2}\left(S^{in} - \pi(D)\right),$$

where $\pi$ is defined by (25). Therefore, if the set $p(S^{in})$ is in the interior of $I(S^{in})$, by Fermat's theorem, any point $D^* \in p(S^{in})$ is a critical point of $P$; i.e., $P'(D^*) = 0$, which is equivalent to $S^{in} = \pi(D^*)$. The proof that $p(S^{in})$ is in the interior of $I(S^{in})$ is the same as the proof that $g(S^{in})$ is in the interior of $I(S^{in})$ given in Appendix A.3.1.

Appendix A.4.2. Proof of Proposition 4

Since $Q(\lambda(a)) = Q(S^{in}) = 0$ and $Q(S) > 0$ for $\lambda(a) < S < S^{in}$, the maximum of $Q$ is attained at a point $S^* \in (\lambda(a), S^{in})$. By Fermat's theorem, $S^*$ is a critical point of $Q$, i.e., $Q'(S^*) = 0$. We have

$$P'(D) = \tfrac{1}{\alpha}Q'(\lambda(\alpha D + a))\lambda'(\alpha D + a)$$

Hence, $Q$ has a maximum at $S^*$ if and only if $P$ has a maximum at $D^* = \frac{\mu(S^*)-a}{\alpha}$. Moreover, $Q'(S) = 0$ if and only if $S^{in} = \rho(S)$, where $\eta$ is defined by (30). From $Q'(S^*) = 0$, it is deduced that $S^{in} = \rho(S^*)$.

Appendix A.4.3. Proof of Lemma 2

If $\mu$ is $\mathcal{C}^2$, so is $\lambda$, and the derivative of $\pi$ is given by

$$\pi'(D) = \alpha\frac{\alpha D+a}{\alpha D+2a}\left(\frac{2(\alpha D+3a)}{\alpha D+2a}\lambda'(\alpha D + a) + \alpha D\lambda''(\alpha D + a)\right).$$

Using (A2), we have

$$\pi'(D) = \alpha\frac{\alpha D+a}{\alpha D+2a}\lambda'(\alpha D + a)\left(\frac{2(\alpha D+3a)}{\alpha D+2a} - \frac{\alpha D\mu''(\lambda(\alpha D+a))}{(\mu'(\lambda(\alpha D+a)))^2}\right).$$

Since $\lambda' > 0$ it is deduced that $\pi'(D) > 0$ if and only if for all $D \in (0, \delta(S^m))$,

$$\alpha D\frac{\alpha D+2a}{\alpha D+3a}\mu''(\lambda(\alpha D + a)) < 2(\mu'(\lambda(\alpha D + a)))^2.$$

Using the change of variable $S = \lambda(\alpha D + a)$, this condition is equivalent to the following: for all $S \in \lambda(a)0, S^m)$,

$$\frac{(\mu(S)-a)(\mu(S)+a)}{\mu(S)+2a}\mu''(S) < 2(\mu'(S))^2.$$

Therefore, (1) $\Leftrightarrow$ (2). The derivative of $\rho$ is given by

$$\rho'(S) = \frac{\mu(S)}{(\mu(S)+a)^2(\mu'(S))^2}\left(2(\mu(S) + 2a)(\mu'(S))^2 - (\mu(S) - a)(\mu(S) + a)\mu''(S)\right).$$

Therefore (2) $\Leftrightarrow$ (3). If $\mu'' < 0$ on $(\lambda(a), S^m)$, then, since $\mu' > 0$ on $(\lambda(a), S^m)$, the condition $(\mu(S) - a)\mu''(S) < (\mu'(S))^2$ is obviously satisfied. Moreover, we have seen in Lemma 1 that if the condition $\left(\frac{1}{\mu - a}\right)''(S) > 0$ holds, then we have

$$(\mu(S) - a)\mu''(S) < 2(\mu'(S))^2.$$

Therefore, we have

$$\frac{(\mu(S) - a)(\mu(S) + a)}{\mu(S) + 2a}\mu''(S) < (\mu(S) - a)\mu''(S) < 2(\mu'(S))^2,$$

which is the condition 2 in the lemma.

*Appendix A.5. Applications to Some Usual Growth Functions*

For simplicity, we restrict our attention to the case where $\alpha = 1$ and $a = 0$. In this case, $D^*(S^{in})$ is obtained by solving Equation (33). One can also solve Equation (32), to get the maximum $S^*(S^{in})$, and then take

$$D^*(S^{in}) = \mu(S^*(S^{in})). \tag{A3}$$

Appendix A.5.1. Monod Growth Rate

This growth function is given by (4). This function satisfies Hypothesis 1 with $S^m = +\infty$. Since $\mu'' < 0$, using Lemma 3, we obtain that Hypothesis 4 is satisfied. Straightforward computations show that

$$\lambda(D) = \frac{DK}{m - D}, \qquad \gamma(D) = \frac{DK(2m - D)}{(m - D)^2}, \qquad \eta(S) = S^2/K + 2S.$$

Hence, $S^*(S^{in})$, the (unique) solution of equation $S^{in} = \eta(S)$, and $D^*(S^{in})$ are given by

$$S^*(S^{in}) = \sqrt{K^2 + KS^{in}} - K, \quad D^*(S^{in}) = \mu(S^*(S^{in})) = m\left(1 - \sqrt{\frac{K}{K + S^{in}}}\right).$$

This formula for $D^*(S^{in})$ is well known in the literature; see for example [12,19,20]. In Figure A1a, we show the OD, together with the set of best operating conditions $\Gamma$ and the biogas flow rate $G(D, S^{in})$, with $S^{in} = 10$, for the Monod growth function (4), with $m = 1$ and $K = 5$. This figure shows how the optimal dilution rate $D^*(S^{in})$ can be graphically determined. Although we have an explicit formula for $D^*(S^{in})$, this graphical construction can be very useful as it allows the dilution rate that the experimenter should choose to optimise the biogas flow rate to be visualised in the OD.

Appendix A.5.2. Hill Growth Rate

This growth function is given by

$$\mu(S) = \frac{mS^p}{K^p + S^p}, \qquad p \geq 1. \tag{A4}$$

This function satisfies Hypothesis 1 with $S^m = +\infty$. Moreover, we have

$$\left(\frac{1}{\mu}\right)''(S) = \frac{p(p+1)K^p}{mS^{p+2}}.$$

Hence, $(1/\mu)'' > 0$, and using Lemma 3, we obtain that Hypothesis 4 is satisfied. Notice that for $p > 1$, the Hill function (A4) is not concave on $(0, +\infty)$. Straightforward computations show that

$$\lambda(D) = \left(\frac{D}{m - D}\right)^{\frac{1}{p}}K, \qquad \gamma(D) = \left(\frac{D}{m - D}\right)^{1/p}\frac{(p+1)m - pD}{p(m - D)}K, \qquad \eta(S) = \frac{K^{-p}S^{p+1} + (p+1)S}{p}.$$

Hence, $S^*(S^{in})$, the (unique) solution of equation $S^{in} = \eta(S)$ is the positive solution of equation

$$K^{-p}S^{p+1} + (p+1)S - pS^{in} = 0.$$

One has explicit formulas for $S^*(S^{in})$ when $p = 1$ (the Monod case) and $p = 2$

$$S^*(S^{in}) = \left(K^2 S^{in} + \sqrt{K^6 + K^4(S^{in})^2}\right)^{1/3} - \frac{K^2}{\left(K^2 S^{in} + \sqrt{K^6 + K^4(S^{in})^2}\right)^{1/3}} \quad \text{if} \quad p = 2.$$

We can deduce also the explicit expression of $D^*(S^{in})$, the (unique) solution to equation $S^{in} = \gamma(D)$ by using (A3). This example illustrates the fact that the second method is much more practicable than the first one, since the direct resolution of equation $S^{in} = \gamma(D)$ is not easy.

In Figure A1b, we show the OD, together with the set of best operating conditions $\Gamma$ and the biogas flow rate $G(D, S^{in})$, with $S^{in} = 10$, for the Hill growth function (A4), with $p = 2$, $m = 1$ and $K = 5$. This figure shows how the optimal dilution rate $D^*(S^{in})$ can be graphically determined. This graphical construction is very useful as it allows the dilution rate that the experimenter should choose to optimise the biogas flow rate to be visualised in the OD. Indeed, the above explicit formula for $S^*(S^{in})$, and hence for $D^*(S^{in})$, is not very informative. Moreover, for $p > 2$, we do not have an explicit formula for $D^*(S^{in})$, whereas the graphical construction can be done for any $p$.

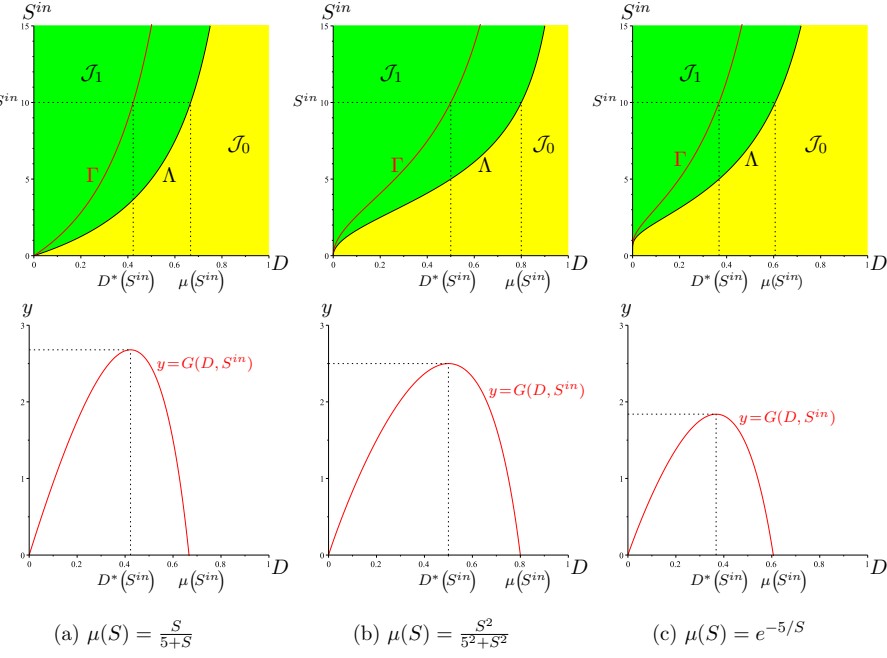

(a) $\mu(S) = \frac{S}{5+S}$      (b) $\mu(S) = \frac{S^2}{5^2+S^2}$      (c) $\mu(S) = e^{-5/S}$

**Figure A1.** The set of best operating conditions $\Gamma$ (in Red) shows the optimal dilution rate $D^*(S^{in})$ for three increasing growth functions and $S^{in} = 10$, $a = 0$, $\alpha = 1$.

Appendix A.5.3. Desmond–Le Quéméner and Bouchez Growth Rate

This growth function is given by [66]

$$\mu(S) = me^{-k/S}. \tag{A5}$$

This function satisfies Hypothesis 1 with $S^m = +\infty$. Moreover, we have

$$\left(\frac{1}{\mu}\right)''(S) = \frac{k}{mS^3}\left(2 + \frac{k}{S}\right)e^{k/S}.$$

Hence, $(1/\mu)'' > 0$, and using Lemma 3, we obtain that Hypothesis 4 is satisfied. Notice that the function (A5) is not concave on $(0, +\infty)$. Straightforward computations show that

$$\lambda(D) = \frac{k}{\ln(m/D)}, \qquad \gamma(D) = \frac{k}{\ln(m/D)}\left(1 + \frac{1}{\ln(m/D)}\right), \qquad \eta(S) = S + \frac{S^2}{k}.$$

Therefore

$$S^*(S^{in}) = \frac{\sqrt{k^2 + 4kS^{in}} - k}{2} \quad \text{and} \quad D^*(S^{in}) = \mu(S^*(S^{in})) = me^{-\frac{\sqrt{k^2 + 4kS^{in}} + k}{2S^{in}}}.$$

In Figure A1c, we show the OD, together with the set of best operating conditions $\Gamma$ and the biogas flow rate $G(D, S^{in})$, with $S^{in} = 10$, for the growth function (A5), with $m = 1$ and $k = 5$. This figure shows how the optimal dilution rate $D^*(S^{in})$ can be graphically determined. Although we have an explicit formula for $D^*(S^{in})$, this graphical construction can be very useful as it allows visualising in the OD the dilution rate that the experimenter chooses to optimise the biogas flow rate.

Appendix A.5.4. Haldane Growth Rate

This growth function is given by (5). It satisfies Hypothesis 1, with

$$S^m = \sqrt{KK_i} \quad \text{and} \quad \max_{S \geq 0} \mu(S) = \mu(S^m) = \frac{m}{1 + 2\sqrt{K/K_i}}.$$

Since $\mu''(S) < 0$ on $(0, S^m)$, using Lemma 3, we obtain that Hypothesis 4 is satisfied. We have

$$\lambda(D) = \frac{m - D - \sqrt{\Delta}}{2D}K_i = \frac{2D}{m - D + \sqrt{\Delta}}K, \qquad \bar{\lambda}(D) = \frac{m - D + \sqrt{\Delta}}{2D}K_i,$$

where $\Delta = (m - D)^2 - 4D^2 K/K_i$, defined for $0 \leq D \leq \mu(S^m)$. Note that $\Delta$ tends toward $(m - D)^2$ when $K_i \to +\infty$. Hence $\lambda(D) \to \frac{DK}{m-D}$ and $\bar{\lambda}(D) \to +\infty$. We find the case of Monod. Straightforward calculations show that

$$\gamma(D) = \frac{2DK(2m - D + 4DK/K_i)}{\Delta + (m - D + 4DK/K_i)\sqrt{\Delta}}, \qquad \eta(S) = \frac{(2K + S)K_i S}{KK_i - S^2}.$$

The solution of $S^{in} = \eta(S)$ is given by

$$S^*(S^{in}) = \frac{\sqrt{KK_i((K + S^{in})K_i + (S^{in})^2)} - KK_i}{K_i + S^{in}}.$$

Hence, $D^*(S^{in})$, the solution of $S^{in} = \gamma(D)$, is given by (A3), i.e.,

$$D^*(S^{in}) = \mu(S^*(S^{in})) = \frac{m(K_i + S^{in})(\sqrt{KK_i((K + S^{in})K_i + (S^{in})^2)} - KK_i)}{2K((K + S^{in})K_i + (S^{in})^2) + (K_i + S^{in} - 2K)\sqrt{KK_i((K + S^{in})K_i + (S^{in})^2)}}.$$

These formulas for $S^*(S^{in})$ and $D^*(S^{in})$ are known in the literature [20]. Note that the equation $S^{in} = \gamma(D)$ is equivalent to an algebraic quadratic equation of degree two which can be solved explicitly. We obtain the formula

$$D^*(S^{in}) = \begin{cases} m\left(\frac{K_i}{K_i - 4K} - \frac{K_i + 2S^{in}}{K_i - 4K}\sqrt{\frac{KK_i}{(K + S^{in})K_i + (S^{in})^2}}\right) & \text{if } K_i \neq 4K, \\ m\frac{S^{in}(4K + S^{in})}{2(2K + S^{in})^2} & \text{if } K_i = 4K. \end{cases}$$

Note that when $K_i \to +\infty$, then $D^*(S^{in}) \to m\left(1 - \sqrt{\frac{K}{K + S^{in}}}\right)$. We find the case of Monod.

On the other hand, equation $\gamma(D) = \bar{\lambda}(D)$ is equivalent to the third-degree polynomial equation:

$$(4K - K_i)^2 D^3 + 3mK_i(4K - K_i)D^2 + 3m^2 K_i(K_i - K)D - m^3 K_i^2 = 0.$$

Therefore $D^c$, considered in Remark 7, is the unique positive solution of this equation and can be computed explicitly. Let us illustrate the results of Section 3.1.4 in the particular case of the Haldane function given by $m = 1$, $K = 5$, and $K_i = 5$. The OD and the set $\Gamma$ of best operating conditions are depicted in Figure A2. The biogas flow is shown for five values of $S^{in}$. The curves $\Gamma$ and $\Lambda_2$ intersect at $(D^c, S^c) = (0.293, 9.397)$. If $S^{in} > S^c$, then the optimal dilution rate $D^*(S^{in})$ corresponds to the bistability region (pink region) $\mathcal{J}_2$. Depending on the initial condition, the system can go to the washout of the species with no biogas production, or its persistence, with maximal biogas production.

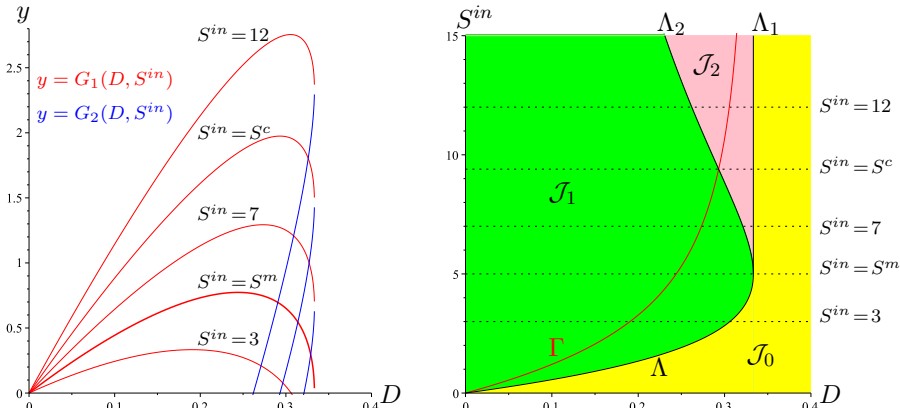

**Figure A2.** The set of optimal biogas production for the Haldane function (5), with $m = 1$, $K = 5$, $K_i = 5$. We have $S^m = 5$, $D^c = 0.293$, and $S^c = 9.397$.

Appendix A.5.5. An Example with Two Maxima

It is known that Hypothesis 1 is not enough to guarantee that the biogas flow rate admits a unique global maximum (Hypothesis 4); see Figure 5.1 in [14]. Even if the function $f$ is increasing, it is possible that the biogas flow rates have two maxima. For example, consider the function

$$\mu(S) = \frac{mS^6 + S}{K^6 + S^6 + S}, \quad \text{with } m = 2, \quad K^6 = 0.1,$$

which is obtained from the Hill function (A4) (with $p = 6$) by adding $S$ to the numerator and denominator. This function is increasing; see Figure A3a. However, for some values of $S^{in}$, the biogas flow rate has three local extrema; see Figure A3d. Numerical exploration shows that the the set of arguments of the maximum of $G$ is as follows

$$g(S^{in}) = \begin{cases} 0.705 & \text{if } S^{in} = 1 \\ \{0.786, 1.277\} & \text{if } S^{in} = 1.7625 \\ 1.475 & \text{if } S^{in} = 2.1 \end{cases}$$

This behaviour is consistent with the plot of the curve $\Gamma$; see Figure A3c. The function $\eta$ is given by:

$$\eta(S) = S + \frac{\mu(S)}{\mu'(S)} = S + \frac{(S + mS^6)(K^6 + S + S^6)}{K^6 + 5(m-1)S^6 + 6K^6mS^5}.$$

The plot of this function shows that it is not increasing; see Figure A3b. Therefore, from Lemma 3, we can easily predict that the function $\gamma$ is not increasing, as depicted in Figure A3c. Hence, Hypothesis 4 is not satisfied.

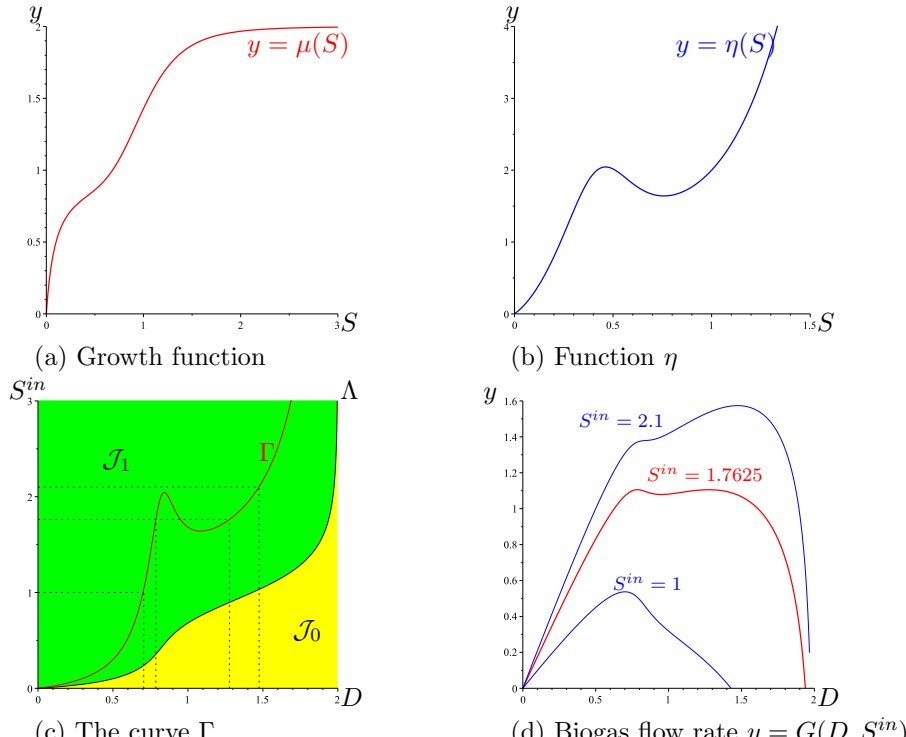

**Figure A3.** An increasing growth function with two maxima of the biogas flow rate.

## Appendix B. Two-Step Models

*Appendix B.1. Model Reduction*

The linear change of variables

$$s_1 = \frac{k_2}{k_1} S_1, \quad x_1 = k_2 X_1, \quad s_2 = S_2, \quad x_2 = k_3 X_2$$

transforms (12) into

$$
\begin{aligned}
\dot{s}_1 &= D\big(s_1^{in} - s_1\big) - f_1(s_1)x_1, \\
\dot{x}_1 &= (f_1(s_1) - D_1)x_1, \\
\dot{s}_2 &= D\big(s_2^{in} - s_2\big) + f_1(s_1)x_1 - f_2(s_2)x_2, \\
\dot{x}_2 &= (f_2(s_2) - D_2)x_2,
\end{aligned}
\tag{A6}
$$

where

$$s_1^{in} = \frac{k_2}{k_1} S_1^{in}, \quad s_2^{in} = S_1^{in}, \quad f_1(s_1) = \mu_1\left(\frac{k_1}{k_2} s_1\right), \quad f_2(s_2) = \mu_2(s_2)$$

Therefore, the stoichiometric coefficients $k_i$, $i = 1, 2, 3$ are reduced to 1. However, as explained in Section 2.2, we do not work with the reduced model (A6) and we present the results in the original model (12).

*Appendix B.2. The Steady States of a Two-Step Model*

The model (12) has a cascade structure, which renders its mathematical analysis easy. There is no additional difficulty compared to the case considered in [26] in which $\alpha_1 = \alpha_2 = \alpha$ and $a_1 = a_2 = 0$. We recall that the break-even concentrations were defined in Table 1. We summarize in Table A2 the definitions of some additional auxiliary functions that are used in the description of the steady states of (12).

**Table A2.** Auxiliary functions. The functions $\lambda_1$, $\lambda_2$, and $\bar{\lambda}_2$ and $H_i$, $i = 1, 2$ are defined in Table 1.

$$S_2^{in*}\left(D, S_1^{in}, S_2^{in}\right) = S_2^{in} + \frac{k_2}{k_1}\left(S_1^{in} - \lambda_1(D_1)\right)$$

$$X_1^*\left(D, S_1^{in}\right) = \frac{D}{k_1 D_1}\left(S_1^{in} - \lambda_1(\alpha D)\right)$$

$$X_{21}\left(D, S_2^{in}\right) = \frac{D}{k_3 D_2}\left(S_2^{in} - \lambda_2(D_2)\right)$$

$$X_{22}\left(D, S_2^{in}\right) = \frac{D}{k_3 D_2}\left(S_2^{in} - \bar{\lambda}_2(D_2)\right)$$

$$X_{2i}^*\left(D, S_1^{in}, S_2^{in}\right) = \frac{D}{k_3 D_2}\left(S_2^{in} + \frac{k_2}{k_1}S_1^{in} - H_i(D)\right), i = 1, 2$$

The system (12) can have up to six steady states, denoted $E_{ij}$, where $i = 0, 1$ and $j = 0, 1, 2$. The convention used is as follows: if $i = 0$, it means that $X_1 = 0$ and if $i = 1$, then $X_1 > 0$. Similarly, if $j = 0$, it means that $X_2 = 0$ and if $j = 1, 2$, then $X_2 > 0$. It should be noticed that $E_{00}$, where $X_1 = 0$ and $X_2 = 0$, is the washout steady state where acidogenic and methanogenic bacteria are extinct; $E_{0i}$, $i = 1, 2$, where $X_1 = 0$ and $X_2 > 0$, is the steady state of washout of acidogenic bacteria, while methanogenic bacteria are maintained; $E_{10}$, where $X_1 > 0$ and $X_2 = 0$ is the steady state of washout of methanogenic bacteria, while acidogenic bacteria are maintained; $E_{1i}$, $i = 1, 2$, where $X_1 > 0$ and $X_2 > 0$ is the steady state of coexistence of acidogenic and methanogenic bacteria. The components of the steady states are given in Table A3.

**Table A3.** The steady states of (12). The functions $\lambda_1$, $\lambda_2$, and $\bar{\lambda}_2$ are defined in Table 1. The functions $S_2^{in*}$, $X_1^*$, $X_{2i}$ and $X_{2i}^*$, $i = 1, 2$ are defined in Table A2.

|  | $S_1$ | $S_2$ | $X_1$ | $X_2$ |
|---|---|---|---|---|
| $E_{00}$ | $S_1^{in}$ | $S_2^{in}$ | 0 | 0 |
| $E_{01}$ | $S_1^{in}$ | $\lambda_2(D_2)$ | 0 | $X_{21}\left(D, S_2^{in}\right)$ |
| $E_{02}$ | $S_1^{in}$ | $\bar{\lambda}_2(D_2)$ | 0 | $X_{22}\left(D, S_2^{in}\right)$ |
| $E_{10}$ | $\lambda_1(D_1)$ | $S_2^{in*}\left(D, S_1^{in}, S_2^{in}\right)$ | $X_1^*\left(D, S_1^{in}\right)$ | 0 |
| $E_{11}$ | $\lambda_1(D_1)$ | $\lambda_2(D_2)$ | $X_1^*\left(D, S_1^{in}\right)$ | $X_{21}^*\left(D, S_1^{in}, S_2^{in}\right)$ |
| $E_{12}$ | $\lambda_1(D_1)$ | $\bar{\lambda}_2(D_2)$ | $X_1^*\left(D, S_1^{in}\right)$ | $X_{22}^*\left(D, S_1^{in}, S_2^{in}\right)$ |

**Table A4.** Necessary and sufficient conditions for the existence and stability of steady states of (12). The functions $\lambda_1$, $\lambda_2$, $\bar{\lambda}_2$, and $H_i$, $i = 1, 2$ are defined in Table 1.

|  | Existence Conditions | Stability Conditions |
|---|---|---|
| $E_{00}$ | Always exists | $S_1^{in} < \lambda_1(D_1)$ and $S_2^{in} \notin \left[\lambda_2(D_2), \bar{\lambda}_2(D_2)\right]$ |
| $E_{01}$ | $S_2^{in} > \lambda_2(D_2)$ | $S_1^{in} < \lambda_1(D_1)$ |
| $E_{02}$ | $S_2^{in} > \bar{\lambda}_2(D_2)$ | Unstable if it exists |
| $E_{10}$ | $S_1^{in} > \lambda_1(D_1)$ | $S_2^{in} + \frac{k_2}{k_1}S_1^{in} \notin [H_1(D), H_2(D)]$ |
| $E_{11}$ | $S_1^{in} > \lambda_1(D_1)$ and $S_2^{in} + \frac{k_2}{k_1}S_1^{in} > H_1(D)$ | Stable if it exists |
| $E_{12}$ | $S_1^{in} > \lambda_1(\alpha D)$ and $S_2^{in} + \frac{k_2}{k_1}S_1^{in} > H_2(D)$ | Unstable if it exists |

**Table A5.** The surfaces $\Lambda_i$, $i = 1 \cdots 6$ and the regions $\mathcal{I}_k$, $k = 0 \cdots 8$.

$$\Lambda_1 = \left\{ (D, S_1^{in}, S_2^{in}) : S_1^{in} = \lambda_1(D_1) := \lambda_1(\alpha_1 D + a_1) \right\}$$
$$\Lambda_2 = \left\{ (D, S_1^{in}, S_2^{in}) : S_2^{in} = \lambda_2(D_2) := \lambda_2(\alpha_2 D + a_2) \right\}$$
$$\Lambda_3 = \left\{ (D, S_1^{in}, S_2^{in}) : S_2^{in} = \bar{\lambda}_2(D_2) := \bar{\lambda}_2(\alpha_2 D + a_2) \right\}$$
$$\Lambda_4 = \left\{ (D, S_1^{in}, S_2^{in}) : S_2^{in} + \tfrac{k_2}{k_1} S_1^{in} = H_1(D) \right\}$$
$$\Lambda_5 = \left\{ (D, S_1^{in}, S_2^{in}) : S_2^{in} + \tfrac{k_2}{k_1} S_1^{in} = H_2(D) \right\}$$
$$\Lambda_6 = \left\{ (D, S_1^{in}, S_2^{in}) : D = \delta_2 := \tfrac{\mu_2(S_2^m) - a_2}{\alpha_2} \right\},$$

$$\mathcal{I}_0 = \left\{ \left( (D, S_1^{in}, S_2^{in}) \right) : S_1^{in} < \lambda_1(D_1) \text{ and } S_2^{in} < \lambda_2(D_2) \right\}$$
$$\mathcal{I}_1 = \left\{ \left( (D, S_1^{in}, S_2^{in}) \right) : S_1^{in} < \lambda_1(D_1) \text{ and } \lambda_2(D_2) < S_2^{in} \leq \bar{\lambda}_2(D_2) \right\}$$
$$\mathcal{I}_2 = \left\{ \left( (D, S_1^{in}, S_2^{in}) \right) : S_1^{in} < \lambda_1(D_2) \text{ and } S_2^{in} > \bar{\lambda}_2(D_2) \right\}$$
$$\mathcal{I}_3 = \left\{ \left( (D, S_1^{in}, S_2^{in}) \right) : S_1^{in} > \lambda_1(D_1) \text{ and } S_2^{in} + \tfrac{k_2}{k_1} S_1^{in} < H_1(D) \right\}$$
$$\mathcal{I}_4 = \left\{ \left( (D, S_1^{in}, S_2^{in}) \right) : S_1^{in} > \lambda_1(D_1), S_2^{in} \leq \lambda_2(D_2) \text{ and } H_1(D) < S_2^{in} + \tfrac{k_2}{k_1} S_1^{in} \leq H_2(D) \right\}$$
$$\mathcal{I}_5 = \left\{ \left( (D, S_1^{in}, S_2^{in}) \right) : S_1^{in} > \lambda_1(D_1), S_2^{in} \leq \lambda_2(D_2) \text{ and } S_2^{in} + \tfrac{k_2}{k_1} S_1^{in} > H_2(D) \right\}$$
$$\mathcal{I}_6 = \left\{ \left( (D, S_1^{in}, S_2^{in}) \right) : S_1^{in} > \lambda_1(D_1), S_2^{in} > \lambda_2(D_2) \text{ and } S_2^{in} + \tfrac{k_2}{k_1} S_1^{in} \leq H_2(D) \right\}$$
$$\mathcal{I}_7 = \left\{ \left( (D, S_1^{in}, S_2^{in}) \right) : S_1^{in} > \lambda_1(D_1), \lambda_2(D_2) < S_2^{in} \leq \bar{\lambda}_2(D_2) \text{ and } S_2^{in} + \tfrac{k_2}{k_1} S_1^{in} > H_2(D) \right\}$$
$$\mathcal{I}_8 = \left\{ \left( (D, S_1^{in}, S_2^{in}) \right) : S_1^{in} > \lambda_1(D_1) \text{ and } S_2^{in} > \bar{\lambda}_2(D_2) \right\}$$

*Appendix B.3. Operating Diagram*

In order to construct the OD of (12), one needs to determine and compute the boundaries of the regions of the diagram, i.e., to compute the parameter values at which a qualitative change in the dynamic behaviour of (12) occurs. These boundaries are six surfaces, denoted $\Lambda_i$, $k = 1 \ldots 6$, in the Set of Operating Parameters (SOP)

$$\text{SOP} = \left\{ (D, S_1^{in}, S_2^{in}) : D \geq 0, S_1^{in} \geq 0 \text{ and } S_2^{in} \geq 0 \right\}.$$

These surfaces separate SOP in nine regions, denoted $\mathcal{I}_k$, $k = 0, \ldots, 8$. These regions correspond to the system behaviour shown in Table A6.

**Table A6.** Existence and stability of steady states of (12) in the nine regions of the operating space. The last column shows the color in which the region is depicted in Figures 5, 6, 8, A4, A5, A7, and A8.

| Region | $E_{00}$ | $E_{01}$ | $E_{02}$ | $E_{10}$ | $E_{11}$ | $E_{12}$ | Color |
|---|---|---|---|---|---|---|---|
| $\mathcal{I}_0$ | GAS | | | | | | Red |
| $\mathcal{I}_1$ | U | GAS | | | | | Blue |
| $\mathcal{I}_2$ | LAS | LAS | U | | | | Cyan |
| $\mathcal{I}_3$ | U | | | GAS | | | Yellow |
| $\mathcal{I}_4$ | U | | | U | GAS | | Green |
| $\mathcal{I}_5$ | U | | | LAS | LAS | U | Pink |
| $\mathcal{I}_6$ | U | U | | U | GAS | | Green |
| $\mathcal{I}_7$ | U | U | | LAS | LAS | U | Pink |
| $\mathcal{I}_8$ | U | U | U | LAS | LAS | U | Pink |

The definitions of the surfaces $\Lambda_i$ and the regions $\mathcal{I}_k$ are given in Table A5. We plot in Figure A6 these surfaces with the biological parameters fixed as in Table A8. Since it is not easy to visualize regions in the three-dimensional operating parameters space, $D$ and $S_1^{in}$ are used as coordinates of the OD, while $S_2^{in}$ is kept constant. The effects of $S_2^{in}$ are shown in a series of operating diagrams; see Figures 5 and A7.

**Remark A1.** *In Figures 5, 6, 8, A4, A5, A7, and A8, presenting ODs, a region is coloured according to the colour in Table A6. Each colour corresponds to different asymptotic behaviour:*

- *Red for the washout of both species; that is, the steady state $E_{00}$ is globally asymptotically stable (GAS), which occurs in region $\mathcal{I}_0$.*
- *Blue for the washout of acidogenic bacteria while methanogenic bacteria are maintained; that is, the steady state $E_{01}$ is GAS, which occurs in region $\mathcal{I}_1$.*
- *Cyan for the bistability of $E_{00}$ and $E_{01}$, which are both (locally) stable. This behaviour occurs in region $\mathcal{I}_1$. Depending on the initial condition, the system can go to the washout of both species or the washout of only the acidogenic bacteria.*
- *Yellow for the washout of methanogenic bacteria while acidogenic bacteria are maintained; that is the steady state $E_{10}$ is GAS, which occurs in region $\mathcal{I}_3$.*
- *Green for the global asymptotic stability of the positive steady state $E_{11}$; which occurs in $\mathcal{I}_4$ and $\mathcal{I}_6$. These regions differ only by the existence, in the second region, of the unstable boundary steady state $E_{01}$.*
- *Pink for the bistability of $E_{10}$ and $E_{11}$, which are both locally asymptotically stable. This behaviour occurs in regions $\mathcal{I}_5$, $\mathcal{I}_7$, and $\mathcal{I}_8$. These regions differ only by the possible existence of the unstable boundary steady states $E_{01}$ or $E_{02}$. Depending on the initial condition, the system can go to the washout of methanogenic bacteria or the coexistence of both species.*

Appendix B.3.1. Operating Diagram in $(S_1^{in}, S_2^{in})$ Where $D$ Is Kept Constant

The fact that there are nine regions is easily seen when considering the sections of SOP through a plane $(S_1^{in}, S_2^{in})$ where $D$ is kept constant. Let us denote

$$\delta_1 = \frac{m_1 - a_1}{\alpha_1}, \quad \delta_2 = \frac{\mu_2(S_2^m) - a_2}{\alpha_2} \tag{A7}$$

The surface $\Lambda_1$ is defined for $D < \delta_1$, the surfaces $\Lambda_2$ and $\Lambda_3$ are defined for $D < \delta_2$, and the surfaces $\Lambda_4$ and $\Lambda_5$ are defined for $D < \min(\delta_1, \delta_2)$, where $\delta_1$ and $\delta_2$ are given by (A7). The intersections of the surfaces $\Lambda_i$, $i = 1 \ldots 5$, with a plane where $D$ is kept constant are straight lines: vertical line for $\Lambda_1$, horizontal lines for $\Lambda_2$ and $\Lambda_3$, and oblique lines for $\Lambda_4$ and $\Lambda_5$; see Figure A4. We consider in this figure the case $\delta_1 > \delta_2$. This case corresponds to the situation where $\alpha_1 = \alpha_2$, $a_1 = a_2$, and

$$\mu_2(S^m) = \max_{S_2 \geq 0} \mu_2(S_2) < \max_{S_1 \geq 0} \mu_1(S_1) = \mu_1(+\infty),$$

which is most likely to occur in a real model. The case $\delta_1 \leq \delta_2$ is similar; see [42]. Since the curves are straight lines, the nine regions of the OD are easy to picture. The regions are coloured according to the colours in Table A6. This table gives the system behaviour in the nine regions.

Figure A4 shows the following features. For $0 < D < \delta_2$, all regions exist; see Figure A4a. For increasing $D$, the vertical line $\Lambda_1$ moves to the right and tends towards the vertical line defined by $S_1^{in} = \lambda_1(\alpha\delta_2 + a_1)$. At the same time, the horizontal lines $\Lambda_2$ and $\Lambda_3$ move towards each other and tend toward the horizontal line defined by $S_2^{in} = S_2^m$, so that the regions $\mathcal{I}_1$, $\mathcal{I}_4$, $\mathcal{I}_6$, and $\mathcal{I}_7$ shrink and disappear; see Figure A4b. For $D = \delta_2$, the OD changes dramatically, since regions $\mathcal{I}_1$, $\mathcal{I}_4$, $\mathcal{I}_6$, and $\mathcal{I}_7$ shrink and disappear; see Figure A4b, obtained for $D < \delta_2$ and $D \approx \delta_2$. For $D > \delta_2$ and $D \approx \delta_2$, regions $\mathcal{I}_0$, $\mathcal{I}_3$ invade the whole operating plane, so that regions $\mathcal{I}_2$, $\mathcal{I}_5$, and $\mathcal{I}_8$ also disappear; see Figure A4c. For $\delta_2 < D < \delta_1$, only regions $\mathcal{I}_0$ and $\mathcal{I}_3$ appear; see Figures A4d. For increasing $D$, the vertical line $\Lambda_1$ moves to the right and tends towards infinity, so that, for $D \geq \delta_1$, only region $\mathcal{I}_0$ appears.

In Figure A4, the axes are not graduated, because the figure corresponds to a general case where the growth functions $\mu_1$ and $\mu_2$ verify Hypotheses 2 and 3 and the biological parameters are not specified. The intersections of the OD with planes where $D$ is constant provide an easy way to see that the OD contains nine regions. However, as we are interested in this paper in the biogas flow rate as a function of $D$, it is preferable to have ODs that

include $D$ as a coordinate and in which, for example, $S_2^{in}$ is fixed. We describe these diagrams in the following section.

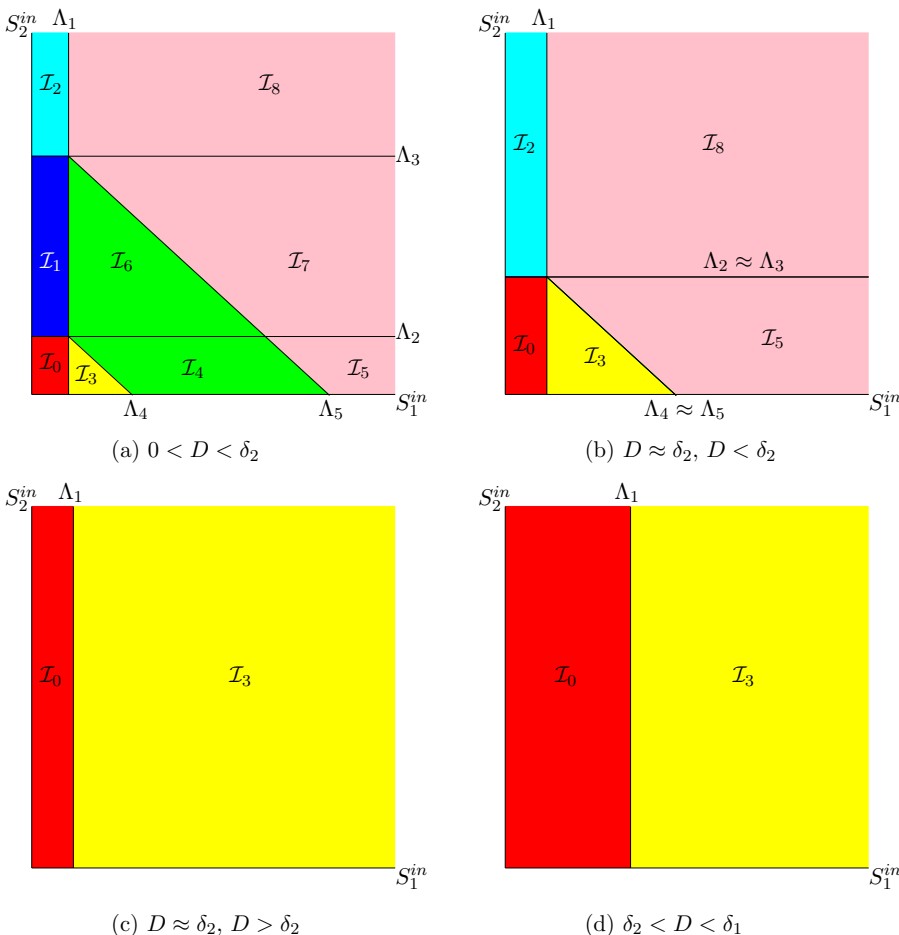

**Figure A4.** The 2-dimensional OD in $\left(S_1^{in}, S_2^{in}\right)$, obtained by cuts at the $D$ constant of the 3-dimensional OD of (12), where $\delta_1$ and $\delta_2$ are given by (A7). If $D \geq \delta_1$, the region $\mathcal{I}_0$ invades the whole plane.

### Appendix B.3.2. Operating Diagram in $(D, S_1^{in})$ Where $S_2^{in}$ Is Kept Constant

Since we want to plot the intersections of the regions $\mathcal{J}_k$ with a $\left(D, S_1^{in}\right)$-plane, where $S_2^{in}$ is kept constant, we must determine the intersections of the surfaces $\Lambda_i$ with this plane. These intersections are the curves whose equations are given in Table A7.

**Table A7.** Intersections of $\Lambda_k$ with a $\left(D, S_1^{in}\right)$-plane, where $S_2^{in}$ is kept constant.

| | |
|---|---|
| $\Lambda_1$ | Curve of function $S_1^{in} = \lambda_1(\alpha_1 D + a_1)$ or $D = \frac{\mu_1(S_1^{in}) - a_1}{\alpha_1}$ |
| $\Lambda_2$ | Vertical line $D = \frac{\mu_2(S_2^{in}) - a_2}{\alpha_2}$ or $S_2^{in} = \lambda_2(\alpha_2 D + a_2)$, if $S_2^{in} \leq S_2^m$ |
| $\Lambda_3$ | Vertical line $D = \frac{\mu_2(S_2^{in}) - a_2}{\alpha_2}$ or $S_2^{in} = \bar{\lambda}_2(\alpha_2 D + a_2)$, if $S_2^{in} \geq S_2^m$ |
| $\Lambda_4$ | Curve of function $S_1^{in} = \frac{k_1}{k_2}\left(H_1(D) - S_2^{in}\right)$ restricted to $S_1^{in} > \lambda_1(\alpha_1 D + a_1)$ |
| $\Lambda_5$ | Curve of function $S_1^{in} = \frac{k_1}{k_2}\left(H_2(D) - S_2^{in}\right)$ restricted to $S_1^{in} > \lambda_1(\alpha_1 D + a_1)$ |
| $\Lambda_6$ | Vertical line $D = \frac{\mu_2(S_2^m) - a_2}{\alpha_2}$ or $S_2^m = \lambda_2(\alpha_2 D + a_2) = \bar{\lambda}_2(\alpha_2 D + a_2)$ |

From the equations of curves $\Lambda_4$ and $\Lambda_4$ and using the $\lambda_2 < \bar{\lambda}_2$, we see that the curve $\Lambda_5$ is above the curve $\Lambda_4$, which is itself above the curve $\Lambda_1$. Note that $\Lambda_1$ and $\Lambda_4$ are increasing, while $\Lambda_5$ is not necessarily increasing, since $H_2(D)$ is the sum of the increasing

function $\frac{k_2}{k_1}\lambda_1(\alpha_1 D + a_1)$ and the decreasing function $\bar{\lambda}_2(\alpha_2 D + a_2)$. In Figure A5, we have depicted the curves in the particular case where the curve $\Lambda_5$ is decreasing. The general case is left to the reader. It is similar to the case (B) and (C) considered in [42].

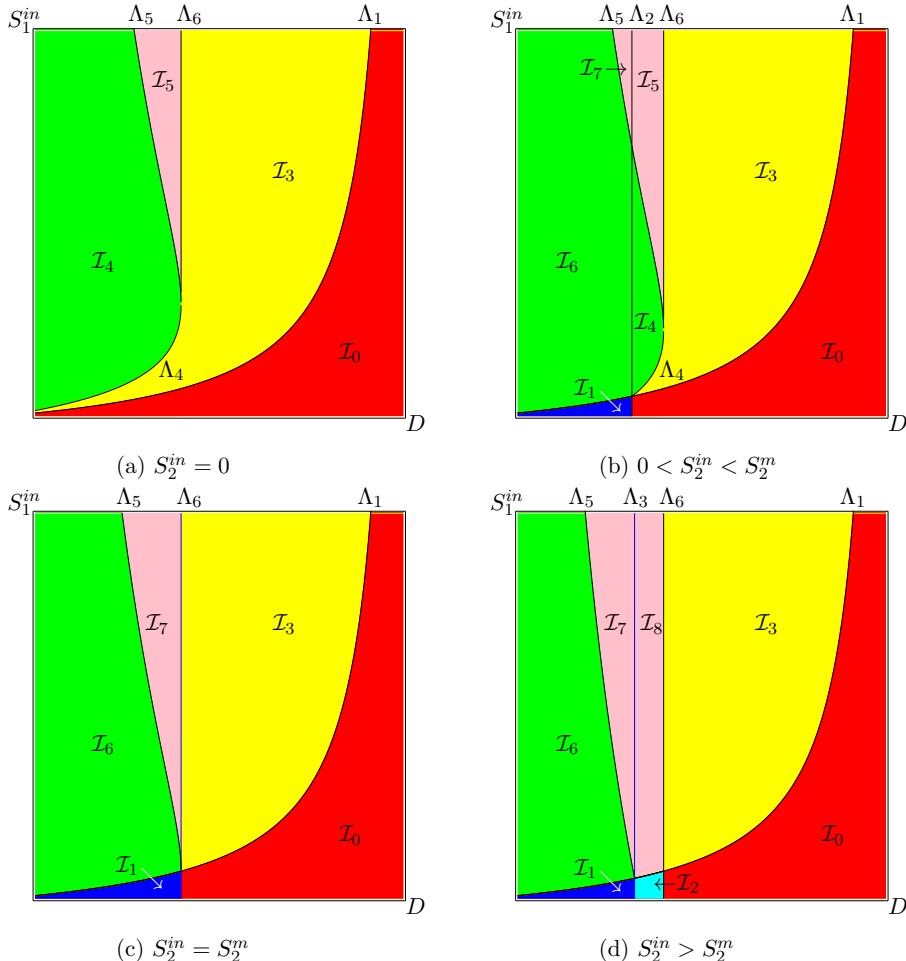

**Figure A5.** The 2-dimensional OD in $\left(D, S_1^{in}\right)$ obtained by cuts at the $S_2^{in}$ constant of the 3-dimensional OD of (12).

From the equations of the curves given in Table A7, we deduce that if $0 \leq S_2^{in} \leq S_2^m$, then curves $\Lambda_4$, $\Lambda_5$ and $\Lambda_6$ intersect at point

$$\Lambda_4 \cap \Lambda_5 \cap \Lambda_6 = \left\{ \left( \delta_2, \frac{k_1}{k_2}(S_2^m - S_2^{in}) + \lambda_1(\alpha_1\delta_2 + a_1) \right) \right\} ,$$

while curves $\Lambda_1$, $\Lambda_2$ and $\Lambda_4$ intersect at point

$$\Lambda_1 \cap \Lambda_2 \cap \Lambda_4 = \left\{ \left( \delta(S_2^{in}), \lambda_1(\alpha_1\delta(S_2^{in}) + a_1) \right) \right\}, \quad \text{where} \quad \delta(S_2^{in}) = \frac{\mu_2(S_2^{in}) - a_2}{\alpha_2};$$

see Figure A5a,b. Similarly, if $S_2^{in} = S_2^m$, then

$$\Lambda_2 = \Lambda_3 = \Lambda_6 \quad \text{and} \quad \Lambda_1 \cap \Lambda_5 \cap \Lambda_6 = \{(\delta_2, \lambda_1(\alpha_1\delta_2 + a_1))\};$$

see Figure A5c, and if $S_2^{in} > S_2^m$, then

$$\Lambda_1 \cap \Lambda_3 \cap \Lambda_5 = \left\{ \left( \delta(S_2^{in}), \lambda_1(\alpha_1\delta(S_2^{in}) + a_1) \right) \right\}, \quad \Lambda_1 \cap \Lambda_6 = \{(\delta_2, \lambda_1(\alpha_1\delta_2 + a_1))\};$$

see Figure A5d. Therefore, the curves intersect as depicted in Figure A5, where the regions are coloured according to the colours in Table A6. This figure shows the following features: For $S_2^{in} = 0$, only the regions $\mathcal{I}_0$, $\mathcal{I}_3$, $\mathcal{I}_4$, and $\mathcal{I}_5$ exist; see Figure A5a. For $0 < S_2^{in} < S_2^m$, the curve $\Lambda_2$ appears, giving birth to $\mathcal{I}_1$, $\mathcal{I}_6$, and $\mathcal{I}_7$ regions; see Figure A5b. For increasing $S_2^{in}$, $\Lambda_4$, and $\Lambda_5$ curves are translated downwards, while the vertical line $\Lambda_2$ moves to the right and tends towards the vertical line $\Lambda_6$, as $S_2^{in}$ tends to $S_2^m$. For $S_2^{in} = S_2^M$, the curve $\Lambda_4$ disappears, while $\Lambda_2$ becomes equal to $\Lambda_6$, so that $\mathcal{I}_4$ and $\mathcal{I}_5$ regions have disappeared; see Figure A5c. For $S_2^{in} > S_2^m$, the curve $\Lambda_3$ appears, giving birth to $\mathcal{I}_2$ and $\mathcal{I}_8$ regions; see Figure A5d. For increasing $S_2^{in}$, the vertical line $\Lambda_3$ moves to the left, while the $\Lambda_5$ curve is translated downwards.

*Appendix B.4. The Operating Diagram to the AM2 Model*

In this section, we show the ODs of the model (50,51), with the biological parameter values given in Table A8. These parameter values can be found in Tables III and V of [25]. These values have been also used by [11]. The OD in the three-dimensional SOP is shown in Figure A6. The two-dimensional diagrams in the $(D, S_1^{in})$ planes where $S_2^{in}$ is kept constant are depicted in Figure A7. The two-dimensional diagrams in the $(S_1^{in}, S_2^{in})$ planes where $D$ is kept constant are depicted in Figure A8.

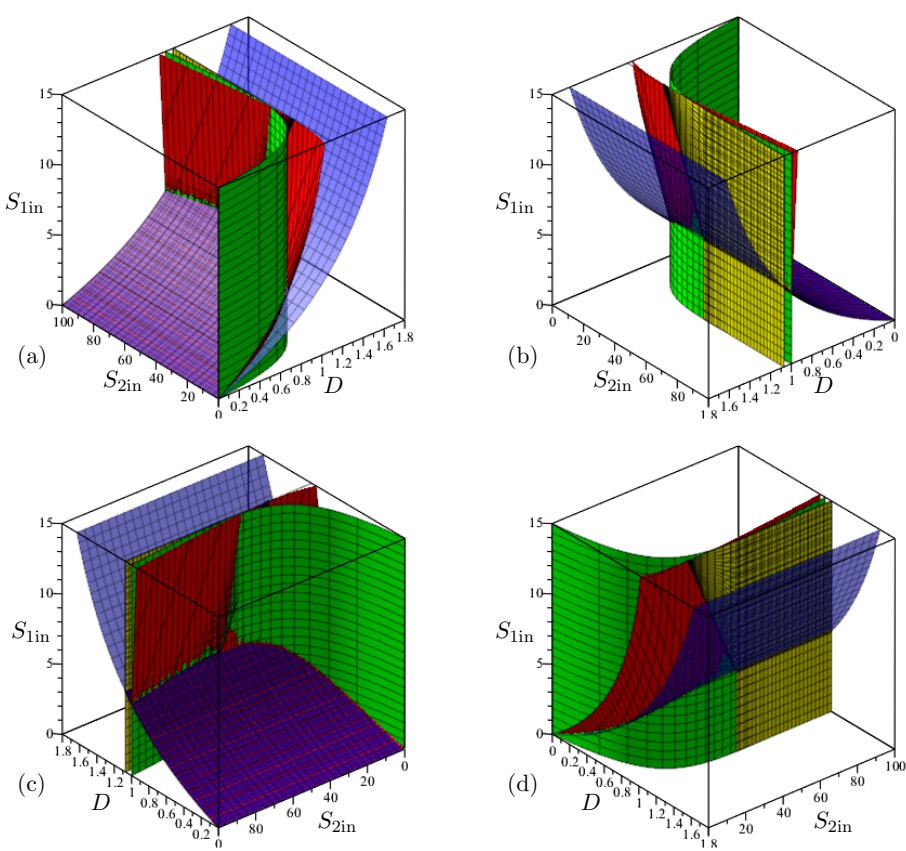

**Figure A6.** The surfaces $\Lambda_1$ (in Blue), $\Lambda_2$ and $\Lambda_3$ (in Green), $\Lambda_4$ and $\Lambda_5$ (in Red), and $\Lambda_6$ (in Yellow), defined in Table A5 separate the 3-dimensional operating space $\left(D, S_1^{in}, S_2^{in}\right)$ in 9 regions $\mathcal{I}_k$, $k = 0, \cdots, 8$. Front (**a**), rear (**b**), left (**c**), and right (**d**) view of the surfaces $\Lambda_i$. The biological parameter values are given in Table A8.

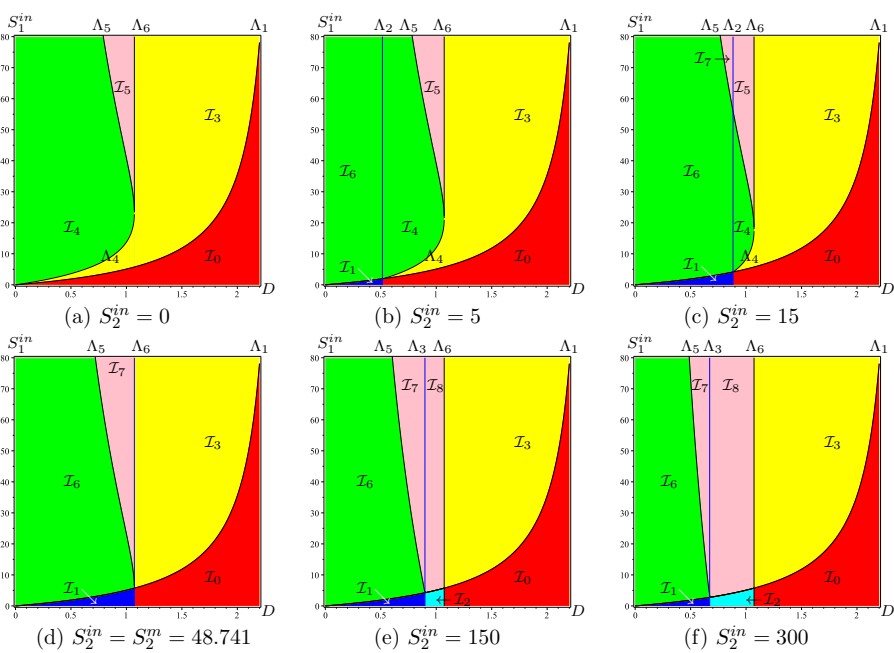

**Figure A7.** The 2-dimensional OD in $\left(D, S_1^{in}\right)$, obtained by cuts at $S_2^{in}$ constant of the 3-dimensional OD shown in Figure A6.

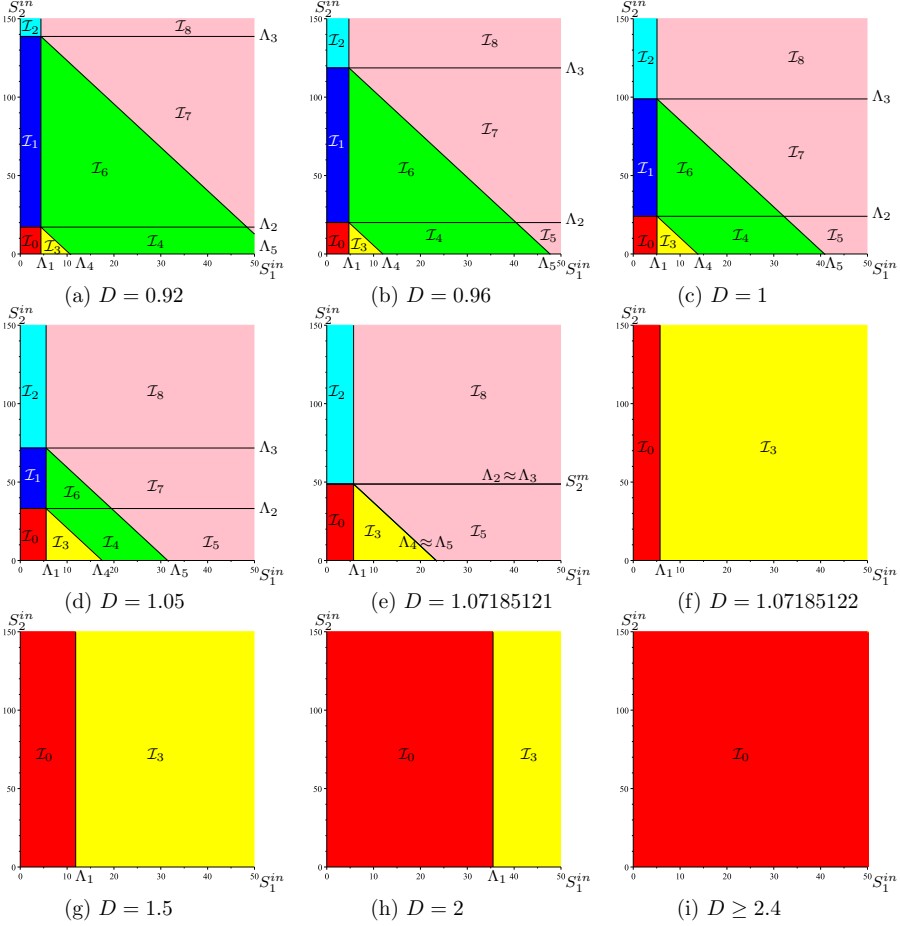

**Figure A8.** The 2-dimensional OD in $\left(S_1^{in}, S_2^{in}\right)$, obtained by cuts at $D$ constant of the 3-dimensional OD shown in Figure A6.

**Table A8.** Nominal parameters values, and $\alpha = 0$, used in Figures 5, 7, 8, A6, A7, and A8.

| Parameter | $m_1$ | $K_1$ | $m_2$ | $K_2$ | $K_I$ | $k_1$ | $k_2$ | $k_3$ |
|---|---|---|---|---|---|---|---|---|
| Unit | $d^{-1}$ | g/L | $d^{-1}$ | mmol/L | mmol/L | | mmol/g | mmol/g |
| Value | 1.2 | 7.1 | 0.74 | 9.28 | 256 | 42.14 | 116.5 | 268 |

*Appendix B.5. Maximization of Biogas Production*

Appendix B.5.1. Proof of Proposition 5

From Table 2, it is seen that $G_{02}$ is defined if and only if $\bar{\lambda}_2(D_2) < S_2^{in}$. Since $\bar{\lambda}_2(D_2) > \lambda_2(D_2)$, the results show that $G_{01}$ is also defined and $G_{01}(D, S_2^{in}) > G_{02}(D, S_2^{in})$. This proves the first item of the proposition.

From Table 2, it is seen that $G_{12}$ is defined if and only if $H_2(D) < S_2^{in} + \frac{k_2}{k_1}S_1^{in}$. Since $H_2(D) > H_1(D)$, the results show that $G_{11}$ is also defined and $G_{11}(D, S_1^{in}, S_2^{in}) > G_{12}(D, S_1^{in}, S_2^{in})$. This proves the second item of the proposition.

If $G_{11}$ is defined, then $S_1^{in} > \lambda_1(D_1)$. Hence,

$$S_2^{in} + \tfrac{k_2}{k_1}S_1^{in} - H_1(D) = S_2^{in} - \lambda_2(D_2) + \tfrac{k_2}{k_1}\left(S_1^{in} - \lambda_1(D_1)\right) > S_2^{in} - \lambda_2(D_2).$$

Therefore, if $G_{01}$ is defined, we have $G_{11}(D, S_1^{in}, S_2^{in}) > G_{01}(D, S_2^{in})$. This proves the third item of the proposition.

Appendix B.5.2. Proof of Proposition 6

The proof follows the same ideas and computations as the proof of Proposition 1. See Appendix A.3.1 for the details.

Appendix B.5.3. Proof of Proposition 7

Since the functions $\gamma_2$ and $\mu_1$ are increasing, the function $S_1^{in} \mapsto \gamma_2\left(\mu_1\left(S_1^{in}\right)\right)$ is increasing. Therefore, the condition $S_1^{in} > S_1^0$ is equivalent to the fact that the point $(S_1^{in}, S_2^{in})$ lies to the right of the curve $\Sigma_0$. Similarly, if the function $\mu_i/\mu_1'$ is increasing, then the function

$$S_1^{in} \mapsto \gamma_2\left(\mu_1\left(S_1^{in}\right)\right) + \frac{k_2}{k_1}\frac{\mu_1\left(S_1^{in}\right)}{\mu_1'\left(S_1^{in}\right)}$$

is increasing. Therefore the condition $S_1^{in} < S_1^1$ is equivalent to the fact that the point $(S_1^{in}, S_2^{in})$ lies to the left of the curve $\Sigma_1$.

Appendix B.5.4. Proof of Proposition 8

Equation (49) is equivalent to the equation

$$G_0(D_0^*(S_2^{in})) = G_1(D_1^*(S_1^{in}, S_2^{in}))$$

where $D_0^*(S_2^{in})$ is the solution to (38) and $D_1^*(S_1^{in}, S_2^{in})$ is the solution to (40). Therefore, using (34) and (35), we deduce that we need to solve the following system of three equations with four unknowns $S_1^{in}$, $S_2^{in}$, $D_0$, and $D_1$.

$$D_0\left(S_2^{in} - \lambda_2(\alpha D_0)\right) = D_1\left(S_2^{in} + \tfrac{k_2}{k_1}S_1^{in} - \lambda_2(\alpha D_1) - \tfrac{k_2}{k_1}\lambda_1(\alpha D_1)\right), \tag{A8}$$

$$S_2^{in} = \gamma_2(\alpha D_0), \tag{A9}$$

$$S_2^{in} + \tfrac{k_2}{k_1}S_1^{in} = \gamma_2(\alpha D_1) + \tfrac{k_2}{k_1}\gamma_1(\alpha D_1). \tag{A10}$$

Substituting (A9) and (A10) into (A8), we obtain

$$D_0(\gamma_2(\alpha D_0) - \lambda_2(\alpha D_0)) = D_1\left(\gamma_2(\alpha D_1) + \tfrac{k_2}{k_1}\gamma_1(\alpha D_1) - \lambda_2(\alpha D_1) - \tfrac{k_2}{k_1}\lambda_1(\alpha D_1)\right).$$

Replacing $\gamma_2$ and $\gamma_1$ by their expressions (36) and (37), respectively, we obtain

$$D_0^2 \lambda_2'(\alpha D_0) = D_1^2\Big(\lambda_2'(\alpha D_1) + \tfrac{k_2}{k_1}\lambda_1'(\alpha D_1)\Big).$$

Therefore, $\alpha D_0$ is a solution to equation

$$\phi(\alpha D_0) = \alpha^2 D_1^2\Big(\lambda_2'(\alpha D_1) + \tfrac{k_2}{k_1}\lambda_1'(\alpha D_1)\Big),$$

where $\phi$ is as in Hypothesis (8). Using this hypothesis, we obtain $\alpha D_0 = \Delta(D_1)$, where $\Delta$ is given by (48). Substituting in (A9) and (A10), we obtain

$$S_2^{in} = \gamma_2(\Delta(D_1)), \qquad \gamma_2(\Delta(D_1)) + \tfrac{k_2}{k_1}S_1^{in} = \gamma_2(\alpha D_1) + \tfrac{k_2}{k_1}\gamma_1(\alpha D_1).$$

These equations show that the point $(S_1^{in}, S_2^{in})$ belongs to the curve $\mathcal{C}$, defined by equations (47). The system formed by the three Equations (A8)–(A10) shows that the reciprocal is also true, i.e., any point on curve $\mathcal{C}$ is a point where $\max_D G_0 = \max_D G_1$. Since the partial derivative of $G_1$ with respect to $S_1^{in}$ is positive, we see that we have $\max_D G_1 > \max_D G_0$ to the right of curve $\mathcal{C}$.

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
