# Peer review of "Best Operating Conditions for Biogas Production in Some Simple Anaerobic Digestion Models"

_processes, doi:10.3390/pr10020258_

Round 1

Reviewer 1 Report

This manuscript reported the best operating conditions for biogas production in some simple anaerobic digestion models. This study may be interesting, but the manuscript in current version does not meet the level for publication.

The Contents of this manuscript should be removed. Generally, there is no Contents for research paper.

The Introduction seems to be redundant, and revisions should be made to refine the manuscript.

As to section “2 Materials and Methods”, key information should be described, and additionally information should be put in Supporting Information.

In the section of “Results”, there are too many figures, some description and figures should be put in Supporting Information.

The description and format of the manuscript should be revised to meet the paper level.

As the authors proposed the models, the application of the models to experimental data or reported data should be conducted to prove the accuracy of the models.

Author Response

Olease see the attachment

Reviewer 2 Report

Manuscript ID processes-1544548 entitled "Best operating conditions for biogas production in some simple anaerobic digestion models" is an interesting attempt to optimize methane fermentation using mathematical models. In my opinion, the importance of using this type of techniques is becoming more and more important and this is another work enriching the existing knowledge. However, I believe that this kind of modeling has very limited relevance for practical application in biogas production systems. In some cases, it does not cover all complex processes and technological parameters of the methane fermentation process. I also believe that this type of analysis is of very limited importance for the operators of agricultural and utilities biogas plants. Nevertheless, I believe that the work is well written and corresponds to the profile of Processes journal. Below are my critical remarks, the inclusion of which will increase the universality and relevance of the content contained in the manuscript:

1. The appropriate and up-to-date template for Journal Processes should be used. Where a lot of important information should be included.

2. Abstract must be completed. Abstract - The author fails to emphasize the novelty and significance of the study. Authors should clearly formulate the aim of the research An abstract summarizes, usually in one paragraph of 200-300 words or less, the major aspects of the entire paper in a prescribed sequence that includes: i) the overall purpose of the study and the research problem (s) you investigated; ii) the basic design of the study; iii) major findings or trends found as a result of your analysis; and, iv) a brief summary of your interpretations, recommendations as a way forward and conclusions.

3. I believe that the best method of selecting the best conditions for biogas production is to carry out multi-variant tests and to select the most effective variants, and then to optimize them and develop a mathenatic model. I believe that in the introduction or discussion, such optimization methods should be presented and related to the presented in the smanuscript. Works of this kind are presented in the literature: https://doi.org/10.1016/j.biombioe.2020.105913, https://doi.org/10.3390/pr9081324, https://doi.org/10.3390/ijerph182211988

4. The author writes in the title "Best operating conditions for biogas production ....". It should be emphasized that the conditions and technological parameters characterizing the methane fermentation process include: hydraulic retention time, organic load rate, concentration of anaerobic sludge in bioreactor, substrate dehydration, organic matter content, substrate dosing, mixing method and frequency, temperature and many , many other. Please specify exactly which ones are selected.

5. The conclusions are too general and do not define the selected conditions of the methane fermentation process. They should be redrafted and completed.

6. There are many technical flaws in the manuscript, eg references are mixed with figures. It needs to be corrected.

7.More new scientific publications should be introduced into the manuscript. Many of the works cited by the authors are outdated and quite old.

Round 2

Reviewer 2 Report

Manuscript has been improved and in my opinion it can be publish in current form.